# Merging by Matching Models in Task Parameter Subspaces

**Derek Tam**                                                                *dtam@cs.toronto.edu*
*University of Toronto*
*Vector Institute*

**Mohit Bansal**                                                             *mbansal@cs.unc.edu*
*University of North Carolina*
*Chapel Hill*

**Colin Raffel**                                                             *craffel@gmail.com*
*University of Toronto*
*Vector Institute*

**Reviewed on OpenReview:** *https://openreview.net/forum?id=qNGo6ghWFB*

## Abstract

Model merging aims to cheaply combine individual task-specific models into a single multi-task model. In this work, we view past merging methods as leveraging different notions of a "task parameter subspace" in which models are matched before being merged. We connect the task parameter subspace of a given model to its loss landscape and formalize how this approach to model merging can be seen as solving a linear system of equations. While past work has generally been limited to linear systems that have a closed-form solution, we consider using the conjugate gradient method to find a solution. We show that using the conjugate gradient method can outperform closed-form solutions, enables merging via linear systems that are otherwise intractable to solve, and flexibly allows choosing from a wide variety of initializations and estimates for the "task parameter subspace". We ultimately demonstrate that our merging framework called "**Ma**tching Models in their **T**ask Parameter **S**ubspace" (`MaTS`) achieves state-of-the-art results in multitask and intermediate-task model merging. We release all of the code and checkpoints used in our work.[1]

## 1 Introduction

The widespread fine-tuning of public pre-trained models has produced a huge number of specialized models. These specialized models may be trained on different tasks, where a "task" is simply the input-output relationship that we aim to train a model to perform (e.g. sentiment analysis of text, object recognition in images, etc.). Alternatively, the Stable Diffusion XL model (Podell et al., 2023) forms the basis of over a thousand specialized image generation models on the Hugging Face Model Hub that are specialized to different styles or content types. How can we recycle these specialized models to create better base models (Choshen et al., 2022; Ramé et al., 2022)? Model merging (Wortsman et al., 2022b; Matena & Raffel, 2022) aims to tackle this problem by combining specialized models into a single model that retains the individual models' capabilities. A common example application of merging is constructing a multitask model from individual-task models, which is the primary application we explore in our paper. Compared to multitask learning, merging does not require simultaneous access to the individual-task datasets. Compared to output-space ensembling of $M$ models, merging produces a model that is $M$ times cheaper to run.

While merging via simple parameter averaging can work well for models that share an architecture and initialization (McMahan et al., 2017; Stich, 2018), recent merging methods improve over simple averaging by

---

[1]https://github.com/r-three/mats

considering parameter importance (Matena & Raffel, 2022), matching activations (Jin et al., 2022), omitting the contribution of the pre-trained model (Ilharco et al., 2022), or resolving interference across models (Yadav et al., 2023).

In our work, we show how several recent merging methods can be viewed as finding a single model that matches task-specific models in their respective "task parameter subspaces". We define a task parameter subspace as the subspace implicitly used by a given merging method that aims to correspond to the important dimensions in parameter space for the task. To match models in their task parameter subspace, merging methods upweight each model in its task parameter subspace, which aims to ensure that the task-relevant components of a given model will not be washed out after the models are combined. In particular, we show how Fisher merging (Matena & Raffel, 2022), RegMean (Jin et al., 2022), and simple parameter averaging (McMahan et al., 2017; Stich, 2018) all perform merging in this way and differ only in their choice of task parameter subspace. Concurrently, other works have focused on inaccuracies in model merging stemming from gradient mismatches in different models, and use these insights to connect diagonal Fisher merging and Task Arithmetic Daheim et al. (2023).

Matching models in their task parameter subspace requires solving a linear system of equations. This linear system implicitly defines a merging objective that relates to a given merging method's choice of task parameter subspace. While previous merging methods used merging objectives with a tractable closed-form solution, we instead develop a merging framework that uses the conjugate gradient method (Hestenes & Stiefel, 1952) to solve a given linear system. We refer to our merging framework as MaTS (**Ma**tching Models in their **T**ask **S**ubspace). By using the conjugate gradient method, MaTS flexibly supports different merging objectives and initializations (which can impact convergence speed). MaTS also enables the use of merging objectives for linear systems that don't have a tractable closed-form solution. To explore this possibility, we leverage insights from K-FAC (Grosse & Martens, 2016; Martens & Grosse, 2015) and introduce a merging method where a model's task parameter subspace is based on a block-diagonal approximation of the Fisher information matrix.

To explore the effectiveness of MaTS, we comprehensively compare it to existing merging methods on multitask and intermediate-task merging of language models and vision models trained via parameter-efficient or full-model fine-tuning. We first explore using preexisting merging methods as an initialization for MaTS and demonstrate that MaTS can significantly boost performance given a suitable choice of merging objective. In particular, in multitask language model merging, we find that MaTS attains state-of-the-art results by a large margin. We use insights from this exploration to develop an effective merging recipe (i.e. a consistent initialization and objective to use) for parameter-efficient and full-model fine-tuning, which we then apply to multitask vision model merging and intermediate-task language model merging. In both cases, we validate that MaTS can boost performance over its initialization and often attains state-of-the-art results. Finally, we discuss how MaTS has a higher computational cost than existing merging methods but is nevertheless dramatically cheaper than explicit multitask training. Taken as a whole, our results validate both our perspective of model merging as matching models in their task parameter subspace as well as the effectiveness of using the conjugate gradient method for solving the corresponding linear system.

## 2 Background

In this paper, we assume we are given $M$ models to merge, all sharing a common architecture and initialization (e.g. a pre-trained model) and denote the parameters of model $m$ as $\boldsymbol{\theta}_m$, the input as $x$, and the output as $y$ over which the model aims to parameterize a distribution $p_{\boldsymbol{\theta}_m}(y|x)$. We further assume we have access to the dataset $D_m$ (containing $N_m$ examples) and the loss function $L_m$ used to train model $m$. All of the models we study use a typical neural network architecture that is made up of a series of linear operations ("layers") with parameter-free nonlinearities in between. Consider linear layer $l$ in some particular model $m$ with weight matrix $\boldsymbol{W}_m \in \mathbb{R}^{d \times k}$. Let $\boldsymbol{z} \in \mathbb{R}^d$ be the layer's input activation, $\boldsymbol{o} \in \mathbb{R}^k$ be the output activation, and $\boldsymbol{o}' \in \mathbb{R}^k$ be the gradient of the loss function with respect to the linear layer's output activation for one particular input. Throughout this work, we assume that models use a cross-entropy loss function, so that $\boldsymbol{s}'$ can be considered as the gradient of the log probability of the correct class with respect to the linear layer's output. For brevity, we refer to this quantity simply as the "output activation gradient". Let $\boldsymbol{Z}_m \in \mathbb{R}^{N_m \times d}$

be the input activations stacked row-wise, $\boldsymbol{O}_m \in \mathbb{R}^{N_m \times k}$ be the output activations stacked row-wise, and $\boldsymbol{O}'_m \in \mathbb{R}^{N_m \times k}$ be the output activation gradients stacked row-wise. The operation computed by the linear layer can therefore be expressed as $\boldsymbol{O}_m = \boldsymbol{Z}_m \boldsymbol{W}_m$. We also allow for a slight abuse of notation and use $\boldsymbol{\theta}_m$ to refer to either the $dk$-dimensional vector of the parameters of a particular linear layer (e.g. by flattening a corresponding matrix) or the $p$-dimensional vector of all parameters in the model. When clear from context, in inline equations we abbreviate summation limits (e.g. $\sum_m$ in place of $\sum_{m=1}^{M}$).

## 2.1   Simple Averaging

A common way to merge models is to simply average their parameters, i.e. compute $\boldsymbol{\theta}^\star = \frac{1}{M} \sum_m \boldsymbol{\theta}_m$ Such parameter averaging has been widely used in federated learning (McMahan et al., 2017) and distributed optimization (Stich, 2018) but has more recently been used to merge models to retain out-of-distribution performance (Wortsman et al., 2022b), improve a pre-trained model (Choshen et al., 2022; Don-Yehiya et al., 2022), or create multimodal models (Sung et al., 2023).

## 2.2   Fisher Merging

Matena & Raffel (2022) view the problem of merging models as finding the set of parameters with the highest joint probability according to the individual model's posteriors:

$$\boldsymbol{\theta}^* = \arg\max_{\boldsymbol{\theta}} \sum_{m=1}^{M} \log P(\boldsymbol{\theta}|D_m) \tag{1}$$

Since maximum likelihood-based training of neural networks does not provide access to a posterior distribution over parameters, the posterior must be approximated. Matena & Raffel (2022) point out that simple averaging solves eq. (1) if parameters are assumed to be sampled from isotropic Gaussian posteriors. However, an isotropic Gaussian is likely a poor approximation of the true posterior, so Matena & Raffel (2022) use the Laplace approximation, which assumes that parameters are sampled from a Gaussian distribution with a mean $\boldsymbol{\theta}_t$ and a covariance set to the inverse of the Fisher information matrix ("Fisher") $\boldsymbol{F}_m$. The Fisher can be generically defined for some model $p_{\boldsymbol{\theta}}(y|x)$ as

$$\boldsymbol{F}_m = \mathop{\mathbb{E}}_{x \sim D_m} \left[ \mathop{\mathbb{E}}_{y \sim p_{\boldsymbol{\theta}}(y|x)} \left[ \nabla_{\boldsymbol{\theta}} \log p_{\boldsymbol{\theta}}(y|x) \nabla_{\boldsymbol{\theta}} \log p_{\boldsymbol{\theta}}(y|x)^\top \right] \right] \tag{2}$$

Since computing, storing, and inverting the Fisher is intractable for modestly large neural networks, Matena & Raffel (2022) use a diagonal approximation of the Fisher $\hat{\boldsymbol{F}}$ (i.e. assuming parameter values are independent), which results in a matrix whose diagonal entries correspond to the the average of the per-example gradients squared. Under this approximation, the parameter values that maximize eq. (1) have the following closed-form solution:

$$\boldsymbol{\theta}^* = \left( 1/\sum_{m=1}^{M} \hat{\boldsymbol{F}}_m \right) \sum_{m=1}^{M} \hat{\boldsymbol{F}}_m \boldsymbol{\theta}_m \tag{3}$$

In practice, it is common to use the empirical Fisher where the expectation over labels is replaced by the ground-truth label for a given datapoint. The empirical Fisher has been shown to have certain limitations compared to the true Fisher when used in the Natural Gradient algorithm (Kunstner et al., 2019). However, we found that for model merging the empirical Fisher worked better (see section 6.6 for full results) and is computationally cheaper and therefore we use it throughout this work.

## 2.3   RegMean

RegMean (Jin et al., 2022) aims to find a merged model that minimizes the distance between the output activations of the original model and the output activations of the merged model. To make this goal tractable, RegMean merges the parameters of each linear layer separately. This per-layer objective can then be formulated as the least squares regression problem

$$\min_{\boldsymbol{W}} \sum_{m=1}^{M} \frac{1}{N_m} ||\boldsymbol{O}_m - \boldsymbol{Z}_m \boldsymbol{W}||_2^2 \tag{4}$$

Jin et al. (2022) solve this least squares regression problem using its closed-form solution:[2]

$$\boldsymbol{W}^* = \left( \sum_{m=1}^{M} \frac{1}{N_m} \boldsymbol{Z}_m^\top \boldsymbol{Z}_m \right)^{-1} \left( \sum_{m=1}^{M} \frac{1}{N_m} (\boldsymbol{Z}_m^\top \boldsymbol{Z}_m) \boldsymbol{W}_m \right) \tag{5}$$

Using RegMean for merging therefore requires computing and inverting the gram matrices $\boldsymbol{Z}_m^\top \boldsymbol{Z}_m$. To ensure invertibility, RegMean scales the nondiagonal terms of the Gram matrix with a constant $\lambda$ (with $0 < \lambda < 1$) whose value can be tuned based on performance on held-out data.

## 3    Matching Individual Models in their Task Parameter Subspace

While simple averaging, diagonal Fisher merging, and RegMean appear to be unrelated in their approach, we now demonstrate that all three methods can be seen as computing a merged model that has the generic form

$$\boldsymbol{\theta}^* = \left( \sum_{m=1}^{M} \boldsymbol{C}_m \right)^{-1} \left( \sum_{m=1}^{M} \boldsymbol{C}_m \boldsymbol{\theta}_m \right) \tag{6}$$

$$= \left( \sum_{m=1}^{M} \boldsymbol{Q}_m \boldsymbol{\Lambda}_m \boldsymbol{Q}_m^\top \right)^{-1} \left( \sum_{m=1}^{M} \boldsymbol{Q}_m \boldsymbol{\Lambda}_m \boldsymbol{Q}_m^\top \boldsymbol{\theta}_m \right) \tag{7}$$

where $\boldsymbol{C}_m$ is a (approximate) covariance matrix of some random variable (i.e. some data). As we will discuss later, prior merging methods primarily differ in terms of the choice of the random variable. Since the covariance matrix is positive semi-definite, we can obtain its eigendecomposition $\boldsymbol{C}_m = \boldsymbol{Q}_m \boldsymbol{\Lambda}_m \boldsymbol{Q}_m^\top$. If $\boldsymbol{C}_m$ is the covariance of some data $\boldsymbol{P}_m$ with SVD decomposition $\boldsymbol{P}_m = \boldsymbol{U}_m \boldsymbol{\Sigma}_m \boldsymbol{V}_m^\top$, then $\boldsymbol{Q}_m = \boldsymbol{V}_m$ as shown in appendix A.3. The top singular vectors in $\boldsymbol{V}_m$ and, equivalently, $\boldsymbol{Q}_m$ are the directions that best capture the variance of $\boldsymbol{P}_m$. In other words, the top column vectors in $\boldsymbol{V}_m$ and $\boldsymbol{Q}_m$ can therefore viewed as forming a basis that is "best aligned" with the data $\boldsymbol{P}_m$. We therefore refer to $\boldsymbol{Q}_m$ as the basis vectors of the "task parameter subspace" whose relative importance are determined by $\boldsymbol{\Lambda}_m$.

### 3.1    The Task Parameter Subspace of Prior Merging Methods

Having established the idea of a task parameter subspace, we now apply our perspective to past merging methods and uncover each method's choice of task parameter subspace.

**Simple Averaging.**    Simple averaging sets $\boldsymbol{C}_m = \boldsymbol{I}$ (the identity matrix) thereby setting $\boldsymbol{Q}_m = \boldsymbol{I}$ and $\boldsymbol{\Lambda}_m = \boldsymbol{I}$. Setting $\boldsymbol{Q}_m = \boldsymbol{I}$ means that merging is performed in the original model's parameter space and setting $\boldsymbol{\Lambda}_m = \boldsymbol{I}$ implies that that each direction in the task parameter subspace is equally important.

**Fisher Merging.**    As established in appendix A.1, the closed-form solution for Fisher merging (i.e. without necessarily making any diagonal approximation to the Fisher) is given by

$$\boldsymbol{\theta}^* = \left( \sum_{m=1}^{M} \boldsymbol{F}_m \right)^{-1} \left( \sum_{m=1}^{M} \boldsymbol{F}_m \boldsymbol{\theta}_m \right) \tag{8}$$

In this case, $\boldsymbol{C}_m = \boldsymbol{F}_m$ is the covariance of the per-example gradients of the datapoints from a task. The true (i.e. not the empirical) Fisher is equal to the Hessian (in expectation) when the negative log-likelihood loss is used (Grosse, 2022a;b; Martens, 2020). In this paper, we only consider the use of the the cross entropy (negative log-likelihood) loss function. While Kunstner et al. (2019) note that the empirical Fisher should go

---

[2]While Jin et al. (2022) do not include the $\frac{1}{N_m}$ in their definitions, the constant $\frac{1}{N_m}$ is used in practice in their implementation and experiments (see https://github.com/bloomberg/dataless-model-merging/blob/main/regmean_demo.ipynb). The constant ensures that each dataset is equally weighed, even if the datasets have different number of examples. Appendix A.2 shows the proof that this modified closed-form solution solves the modified objective.

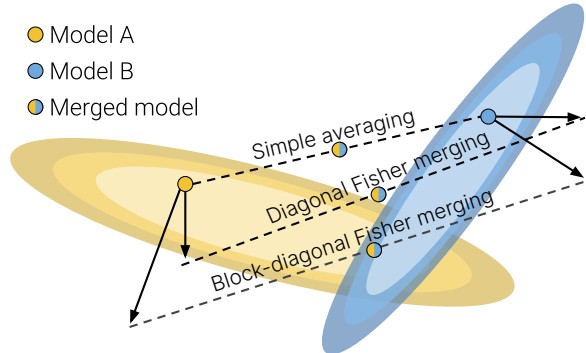

Figure 1: Diagram visualizing what several merging methods are doing in the loss landscape for a particular layer of the model. The ellipses represent the level sets of the losses for different task. Different merging methods modify the individual models in various ways before summing the modified models (with an appropriate normalization constant). Averaging does not modify the model in any way. Diagonal Fisher merging modifies each model by amplifying each parameter along an axis-aligned projection. Block-diagonal Fisher merging modifies each model by amplifying each parameter along a projection along the steepest directions of the loss landscape.

to zero as the model converges whereas the true Fisher doesn't, we follow the insight from Singh & Alistarh (2020) that gradients do not actually vanish for well-optimized neural networks. Thus, we view the (empirical or true) Fisher as approximating the Hessian and therefore $\boldsymbol{Q}_m$ can be viewed as the eigenvectors of the Hessian and $\boldsymbol{\Lambda}_m$ are the corresponding eigenvalues. The Hessian captures the curvature of the model's log-likelihood (Kristiadi, 2018; Martens, 2020), so its top eigenvectors capture the directions of sharpest change in the loss landscape and the bottom eigenvectors capture directions where the loss function is roughly flat (Maddox et al., 2020). As we will discuss in section 3.2, Fisher merging therefore encourages merging to shift models along the given model's parameters "flat" directions.

**Diagonal Fisher Merging.** Diagonal Fisher merging (Matena & Raffel, 2022) approximates $\boldsymbol{C}_m$ as a diagonal Fisher matrix. This corresponds to assuming that $\boldsymbol{Q}_m$ is the identity matrix $\boldsymbol{I}$, which is equivalent to computing the importance score $\boldsymbol{\Lambda}_m$ in axis-aligned parameter space.

**RegMean.** Inspecting eq. (5) suggests that RegMean sets $\boldsymbol{C}_m$ as a function of the scaled Gram matrix $\frac{1}{N_m}\boldsymbol{Z}_m^\top\boldsymbol{Z}_m$, i.e. the covariance of the input activations. Since eq. (5) is defined in terms of a weight matrix and eq. (6) is defined in terms of a parameter vector, RegMean sets $\boldsymbol{C}_m$ as a block-diagonal matrix whose blocks are repeated instances of the scaled Gram matrix $\frac{1}{N_m}\boldsymbol{Z}_m^\top\boldsymbol{Z}_m$. RegMean therefore computes the basis $\boldsymbol{Q}_m$ and important score $\boldsymbol{\Lambda}_m$ based on the directions that best capture the variance in the input activations.

### 3.2 Matching in Task Parameter Subspace

Recall the closed-form solution as written in eq. (6):

$$\boldsymbol{\theta}^* = \underbrace{\left(\sum_{m=1}^{M} \boldsymbol{Q}_m\boldsymbol{\Lambda}_m\boldsymbol{Q}_m^\top\right)^{-1}}_{1}\underbrace{\left(\sum_{m=1}^{M} \boldsymbol{Q}_m\boldsymbol{\Lambda}_m\boldsymbol{Q}_m^\top\boldsymbol{\theta}_m\right)}_{2} \tag{9}$$

The terms of the second summation can be seen as projecting the parameters $\boldsymbol{\theta}_m$ using $\boldsymbol{Q}_m\boldsymbol{\Lambda}_m\boldsymbol{Q}_m^\top$. What does the projection with $\boldsymbol{Q}_m\boldsymbol{\Lambda}_m\boldsymbol{Q}_m^\top$ imply?

1. $\boldsymbol{Q}_m^\top\boldsymbol{\theta}_m$ rotates $\boldsymbol{\theta}_m$ into the task parameter subspace whose basis vectors are the eigenvectors contained in $\boldsymbol{Q}_m$.

2. $\boldsymbol{\Lambda}_m\boldsymbol{Q}_m^\top\boldsymbol{\theta}_m$ amplifies the the dimensions in this rotated coordinate system based on their importance as defined by the eigenvalues.

3. $\boldsymbol{Q}_m\boldsymbol{\Lambda}_m\boldsymbol{Q}_m^\top\boldsymbol{\theta}_m$ rotates back into the original parameter space

The first summation in eq. (9) can then be seen as a normalizing constant to account for all modifications done in the second term. Overall, eq. (9), when $Q_m\Lambda_mQ_m^T$ is the decomposition of a covariance matrix, can be seen as upweighting the "important" components of the model as measured in the task parameter

subspace so that the important components don't get washed out during merging. Thus, the least important component of the models will be shifted more during merging. The components are not constrained to be individual parameters but can be any direction in parameter space. Figure 1 combines this insight with the discussion from section 3.1 and provides a visualization of the different task parameter subspaces implicit in prior merging methods.

Additionally, note that we can re-write the closed-form solution eq. (6) as finding $\boldsymbol{\theta^*}$ such that

$$\sum_{m=1}^{M} \boldsymbol{Q}_m \Lambda_m \boldsymbol{Q}_m^\top \boldsymbol{\theta^*} = \sum_{m=1}^{M} \boldsymbol{Q}_m \Lambda_m \boldsymbol{Q}_m^\top \boldsymbol{\theta}_m \tag{10}$$

One way to satisfy eq. (10) would be if $\boldsymbol{Q}_m \Lambda_m \boldsymbol{Q}_m^\top \boldsymbol{\theta^*}$ matched $\boldsymbol{Q}_m \Lambda_m \boldsymbol{Q}_m^\top \boldsymbol{\theta}_m$ across all tasks $m$. In other words, the extent to which a solution $\boldsymbol{\theta^*}$ satisfies eq. (10) depends on how close $\boldsymbol{Q}_m \Lambda_m \boldsymbol{Q}_m^\top \boldsymbol{\theta^*}$ is to each $\boldsymbol{Q}_m \Lambda_m \boldsymbol{Q}_m^\top \boldsymbol{\theta}_m$. Note that this does not measure the closeness of $\boldsymbol{\theta^*}$ and $\boldsymbol{\theta}_m$ directly, but rather through the projection $\boldsymbol{Q}_m \Lambda_m \boldsymbol{Q}_m^\top$, which (as discussed above) effectively measures closeness in the task parameter subspace.

## 4 Merging via the Conjugate Gradient Method

Merging models by solving for $\boldsymbol{\theta^*}$ in eq. (6) is equivalent to solving a linear system of the form $\boldsymbol{Ax} = \boldsymbol{b}$, where $\boldsymbol{A} = \sum_m \boldsymbol{C}_m$ and $\boldsymbol{b} = \sum_m \boldsymbol{C}_m \boldsymbol{\theta}_m$. Different ways to compute the task parameter subspace define different merging objectives for this linear system. All past merging methods discussed in section 2 provide a tractable closed-form solution to this linear system. However, there may be performant merging methods that do not have such a closed-form solution. In addition, since the solution in eq. (6) involves a matrix inversion, some past methods deviate from the true solution to ensure invertibility. In this work, we therefore explore finding $\boldsymbol{\theta^*}$ via optimization rather than a closed-form solution. Specifically, to solve the linear system, we use the conjugate gradient method (Hestenes & Stiefel, 1952) (CG), a first-order iterative optimization procedure for solving linear systems when $\boldsymbol{A}$ is positive semi-definite (Hayami, 2018). We include more details of CG in appendix B. Note that for eq. (6) $\boldsymbol{A}$ is positive semi-definite since it is the sum of the covariance matrices computed on some data.

A notable benefit of the conjugate gradient method is that it allows for solving a linear system without having to compute a matrix inverse or to compute $\boldsymbol{A}$ explicitly. Specifically, the step size and the update direction can be computed as long as $\boldsymbol{Ax}$ and $\boldsymbol{b}$ can be computed. This is particularly useful if $\boldsymbol{A}$ is too large and/or cannot be explicitly computed. In addition, avoiding the matrix inverse can make CG more stable than a closed-form solution that relies on an inverse. For example, RegMean Jin et al. (2022) has to scale the non-diagonal entries of the gram matrix by a constant to ensure the inverse is not ill-conditioned; we show later in section 6 that CG can find a performant solution to the RegMean objective without applying this scaling. On the whole, we refer to our framework for **ma**tching models in their **t**ask **s**ubspace using conjugate gradient as `MaTS`.

## 5 Block-Diagonal Fisher Merging

Since CG allows merging via linear systems that do not have a tractable closed-form solution, we now introduce a new Fisher merging method that makes use of an improved approximation to the Fisher. Specifically, we follow K-FAC (Grosse & Martens, 2016; Martens & Grosse, 2015) and approximate the Fisher as a block-diagonal matrix (BFM). The closed-form solution of the resulting objective is intractable but can be efficiently optimized with CG.

### 5.1 Connecting Fisher Merging and RegMean

To motivate our use of a block-diagonal approximation to the Fisher, we first note that RegMean can be seen as a form of Fisher merging under certain assumptions. Specifically, RegMean assumes the output neurons are independent and that the gradient of the output activation is constant. Under this assumption,

for a particular linear layer $\boldsymbol{W}^l \in \mathbb{R}^{d \times k}$, the Fisher matrix $\boldsymbol{F}^l \in \mathbb{R}^{dk \times dk}$ is block diagonal matrix with $k$ blocks, each of size $d \times d$. The gradient of a linear layer is the outer product of the input activation and the output activation gradient: $\nabla_{\boldsymbol{W}} \log p_{\boldsymbol{\theta}}(y|x) = \boldsymbol{z} \boldsymbol{o}'^{\top}$. Assuming the output activation gradient $\boldsymbol{o}'$ is 1, for some particular dataset with $N$ examples the $i$th block of a given per-layer Fisher $\boldsymbol{F}^l$ can be shown (in appendix A.4) to be given by $\frac{1}{N} \boldsymbol{Z} \boldsymbol{Z}^{\top}$. Replacing $\boldsymbol{F}_m$ in the Fisher merging objective from eq. (6) with the RegMean-specific Fisher derived gives the RegMean closed-form solution from eq. (5). Following Grosse et al. (2023), this also assumes the gradients of the tokens within an example are independent.

## 5.2  Using K-FAC for Fisher Merging

Can we approximate the Fisher as a block-diagonal matrix without assuming that the output activation gradient is constant? In general, the closed-form solution for Fisher merging with a block-diagonal Fisher approximation would not be tractable to compute for typical neural networks because, for an activation dimensionality of 1024, the Fisher for a single linear layer (i.e. a block in the block-diagonal Fisher approximation) would have 1 trillion entries. One possibility would be to use the K-FAC approximation of the Fisher Martens & Grosse (2015); Grosse & Martens (2016), which approximates the Fisher of a linear layer as the Kronecker product of the covariance of the input activations and the covariance of the output activation gradients:

$$\boldsymbol{F}_m = \left( \frac{1}{N_m} \boldsymbol{Z}_m^{\top} \boldsymbol{Z}_m \right) \otimes \left( \frac{1}{N_m} \boldsymbol{O}_m'^{\top} \boldsymbol{O}_m' \right) \tag{11}$$

where $\otimes$ is Kronecker product.

However, even if we use the K-FAC Martens & Grosse (2015); Grosse & Martens (2016) approximation for the Fisher, a closed-form solution to eq. (6) would still not be tractable. Plugging in the K-FAC approximation into eq. (6) in appendix A.5, we see that

$$\boldsymbol{W}^* = \left[ \sum_{m=1}^{M} \left( \frac{1}{N_m} \boldsymbol{Z}_t^{\top} \boldsymbol{Z}_t \right) \otimes \left( \frac{1}{N_m} \boldsymbol{O}_t'^{\top} \boldsymbol{O}_t' \right) \right]^{-1} \left[ \sum_{m=1}^{M} \left( \frac{1}{N_m} \boldsymbol{Z}_t^{\top} \boldsymbol{Z}_t \right) \boldsymbol{W}_t \left( \frac{1}{N_m} \boldsymbol{O}_t'^{\top} \boldsymbol{O}_t' \right) \right] \tag{12}$$

Though the second term can be easily computed, the first term cannot be since the sum of Kronecker products is not equal to the Kronecker product of sums. While we could assume the sum of Kronecker products is the Kronecker product of sums (an assumption made in (Grosse & Martens, 2016; Martens & Grosse, 2015) to use the K-FAC approximation for optimization), we found in preliminary experiments that this would cause the merged model's performance to crash.

The conjugate gradient method, however, can be used to solve the Fisher merging objective with a block-diagonal Fisher approximation. Specifically, using the K-FAC approximation for the Fisher, we can efficiently compute the matrix vector product $\boldsymbol{Ax}$ (as shown in appendix A.6), even if $\boldsymbol{A}$ cannot be computed with:

$$\boldsymbol{Ax} = \sum_{m=1}^{M} \left( \frac{1}{N_m} \boldsymbol{Z}_t^{\top} \boldsymbol{Z}_t \right) \boldsymbol{W} \left( \frac{1}{N_m} \boldsymbol{O}_t'^{\top} \boldsymbol{O}_t' \right) \tag{13}$$

$$\boldsymbol{b} = \sum_{m=1}^{M} \left( \frac{1}{N_m} \boldsymbol{Z}_t^{\top} \boldsymbol{Z}_t \right) \boldsymbol{W}_t \left( \frac{1}{N_m} \boldsymbol{O}_t'^{\top} \boldsymbol{O}_t' \right) \tag{14}$$

where $\boldsymbol{x} = \text{vec}(\boldsymbol{W})$. Each term in the summands above involve multiplying matrices whose dimensions correspond to the activation dimensions of the model, thereby incurring a modest computational cost. As a result, the use of CG not only avoids an intractable inverse but also produces a computationally feasible method for Fisher merging with a block-diagonal Fisher approximation.

## 6  Experiments

Now that we have introduced `MaTS`, we empirically explore various design choices and validate the effectiveness of our approach. Specifically, we first consider different initializations and objectives in order to identify an

effective fixed recipe to apply in various settings. We mainly focus on merging task-specific models into a single multitask model, but we also consider intermediate-task training in section 6.4. To ensure our method generalizes well to different settings, we perform experiments on both language models and vision models as well as both full-model and parameter-efficient fine-tuning with $(IA)^3$ (Liu et al., 2022), which inserts a trainable vector into specific linear layers.

## 6.1 Baselines

In our experiments, we compare `MaTS` against the following baseline merging methods:

**Simple Averaging** averages the parameter values of the models being merged.

**Ensembling** ensembles the predicted probabilities from the model to merge. Note that this does not result in 1 model, but in $M$ models.

**Task Arithmetic (Ilharco et al., 2022)** introduces a task vector for a particular task as the difference between the fine-tuned parameters (found after training on the task) and the original pre-trained parameters. To construct the merged model, Task Arithmetic adds the scaled sum of the task vectors for all tasks to the pretrained model's parameters. Task Arithmetic introduces the hyperparameter $\lambda$ to rescale the summed task vectors.

**TIES-Merging (Yadav et al., 2023)** improves upon task arithmetic by removing interference between the task vectors. Specifically, TIES zeros out entries in a given task vector that have low magnitude and resolves sign conflicts across different task vectors. Similar to Task Arithmetic, TIES-Merging also uses a hyperparameter $\lambda$ to scale the task vectors after removing interference.

**Diagonal Fisher Merging (Matena & Raffel, 2022)** performs Fisher Merging with a diagonal approximation to the Fisher, as described in section 2.2). To better distinguish between different approximations of the Fisher, we call this method Diagonal Fisher Merging (DFM).

**RegMean (Jin et al., 2022)** solves a linear system to match activations between the merged model and the original models. In practice, RegMean scales the non-diagonal terms of the Gram matrix with a hyperparameter $\lambda$ for numerical stability when computing the inverse. Note that RegMean can only be applied to linear layers since a closed-form solution does not exist otherwise.

**Multitask** training (where a model is trained on all the tasks jointly) is included wherever appropriate and can be considered as a loose upper bound on the performance of a merged multitask model.

We also experimented with Tangent Task Vectors (Ortiz-Jimenez et al., 2023) but were unable to attain reasonable performance with it and therefore did not include it in our results.

We compute the empirical Fisher over the validation set and compare various ways of computing the Fisher in section 6.6. The only hyperparameter in `MaTS` is the number of iterations to run the conjugate gradient method, which we allow to take on values ranging from 10 to 100 in step sizes of 10. We tune the hyperparameters for each method based on validation set performance.

## 6.2 Conjugate Gradient Initializations and Objectives

As discussed in section 3, our use of CG in `MaTS` allows us to consider objectives that various merging methods (implicitly) aim to optimize. In addition, since CG is an iterative optimization method, `MaTS` can flexibly support different choices of an initialization for optimization. While solving the linear system corresponding to eq. (6) is a convex optimization problem, the choice of initialization will effect convergence speed which is of particular importance because we only run CG for at most 100 iterations. We therefore consider using different merging methods (including those that do not fit our framework) to produce an initialization for `MaTS`.

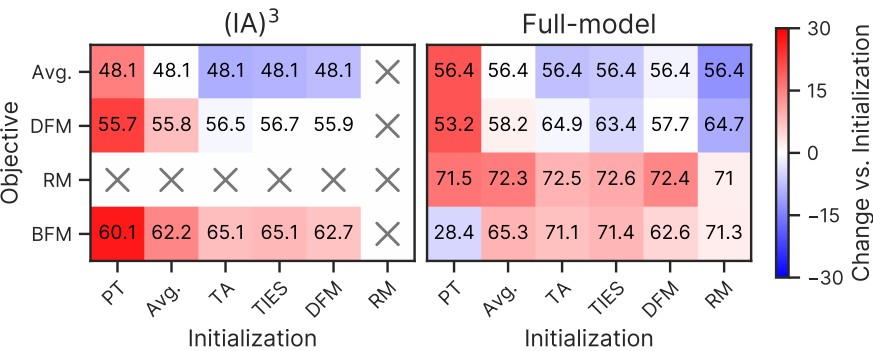

Figure 2: We show the performance using CG to optimize the objectives implicitly defined by various merging methods with various initializations. The number indicates the raw performance and the subscript (and color) indicate the change in performance relative to the initialization.

**Experimental set-up.** As an experimental setting, we focus on merging models fine-tuned on datasets from the T0 mixture (Sanh et al., 2021) to form a multitask models. Zhou et al. (2022) found eight datasets (listed in appendix D.2) were the most important for performance and thus we first focus on merging models fine-tuned on these eight datasets. We compare merging models in the full model fine-tuning setting and in the parameter-efficient fine-tuning setting with $(IA)^3$. We use T5-Large-LM-Adapt (Lester et al., 2021), a variant of T5-Large (Raffel et al., 2020) that was further trained with a language modeling objective as the pre-trained model and include more details on fine-tuning the per-task models in appendix D.1.

**Compatibility considerations.** In this experiment, we consider every possible initialization/objective combination for `MaTS` with the following exceptions: First, when merging $(IA)^3$-trained models, we cannot use RegMean since RegMean is a method for merging linear layers and $(IA)^3$ adds vectors that are elementwise multiplied with activations and does not train any linear layers. We therefore do not consider RegMean as an initialization or objective for models trained with $(IA)^3$. Second, in the full-model fine-tuning setting, we perform BFM using the K-FAC (Grosse & Martens, 2016) approximation as described in section 4 and apply RegMean as usual. However, when merging models fine-tuned with $(IA)^3$, we can perform BFM without resorting to an approximation since the vectors trained by $(IA)^3$ are sufficiently small to perform BFM exactly.

**Results.** In fig. 2, we show the raw performance and performance gain over initialization using `MaTS` with every possible initialization/objective combination. We see that `MaTS` often significantly improves performance compared to a given initialization, except when a relatively poor objective is applied to an effective initialization (e.g. using the "simple averaging" objective with the Task Arithmetic initialization). Notably, we found that using RegMean as both an initialization and objective for `MaTS` led to an improvement in performance over the RegMean-based initialization. This performance boost likely comes from the fact that the matrix inverse needed in RegMean is numerically unstable and so the closed-form solution does not actually result in the optimal solution. To confirm this possibility, we found that using the conjugate gradient method to optimize the RegMean objective achieves a lower objective value than the closed-form solution.

Based on the results in fig. 2, we now design a consistent recipe to apply in the experiments in the rest of the paper. For initialization, we use the Task Arithmetic method due to its strong performance and efficiency in both parameter-efficient and full-model fine-tuning. For the merging objective, in the $(IA)^3$ setup, we use the block-diagonal Fisher merging objective and in the full model fine-tuning setup we use the RegMean objective.

### 6.3 Multitask Models

In this section, we compare the recipe for `MaTS` that we designed in the previous experiment to various other baseline methods on the problem of merging task-specific models to create a single a multitask model in two settings for language and one setting for vision. For language, in the first setting, we use the same

| Domain | Language | | Language | | Vision | |
| --- | --- | --- | --- | --- | --- | --- |
| Tasks | Zhou et al. (2022) | | Yadav et al. (2023) | | Ilharco et al. (2022) | |
| Model | $(IA)^3$ | Full Model | $(IA)^3$ | Full Model | ViT-B/32 | ViT-L/14 |
| Simple Averaging | 48.1 | 56.4 | 52.1 | 60.5 | 65.8 | 79.5 |
| Task Arithmetic | 57.6 | 63.8 | 67.1 | 71.9 | 70.1 | 84.4 |
| TIES-Merging | 56.8 | 62.8 | 66.0 | 69.6 | 73.6 | 85.9 |
| Diagonal Fisher merging | 55.9 | 57.7 | 61.4 | 64.0 | 66.3 | 82.5 |
| RegMean | – | 69.1 | – | 78.9 | 82.0 | 89.6 |
| MaTS | **65.1** | **72.5** | **75.3** | **81.5** | **82.6** | **90.2** |
| Ensemble | 62.8 | 64.9 | 74.8 | 74.1 | 74.0 | 80.0 |
| Multitask (UPPER BOUND) | 76.7 | 80.5 | 82.2 | 84.5 | 89.0 | 93.5 |
| Individual Models | 78.2 | 80.7 | 86.0 | 85.9 | 90.5 | 94.2 |

Table 1: Performance (average accuracy across tasks) of various merging methods and a multitask training-based baseline. The highest score (excluding the multitask performance) is shown in **bold**.

eight key tasks from the T0 training mixture as done in section 6.2 and additionally consider merging fine-tuned variants of T5-Large-LM-Adapt models on the seven tasks used by Yadav et al. (2023) (listed in appendix D.3). For vision, we follow Ilharco et al. (2022); Yadav et al. (2023) and merge the same set of CLIP-based models (Radford et al., 2021) fine-tuned on eight tasks listed in appendix D.4.[3] We present the average test-set performance across all tasks in each mixture attained by each merging method in table 1. Notably, MaTS outperforms all other methods, often by a significant margin. The gains are most pronounced when merging $(IA)^3$-based models in the language domain, where MaTS outperforms prior methods by about 8% absolute. However, we note that none of the methods (including MaTS) match the multitask training-based baseline, suggesting that there still remains room for improvement when merging many individual-task models to create a multitask model. Per-task performance is reported in appendix F.

## 6.4 Intermediate-Task Training

| Intermediate task | MNLI | | QNLI | | QQP | |
| --- | --- | --- | --- | --- | --- | --- |
| Fine-tuning method | $(IA)^3$ | Full Model | $(IA)^3$ | Full Model | $(IA)^3$ | Full Model |
| RTE (BASELINE) | 79.8 | 75.5 | 79.8 | 75.5 | **79.8** | 75.5 |
| Simple Averaging | 79.8 | 78.3 | 79.8 | 75.1 | **79.8** | **78.3** |
| Task Arithmetic | 79.8 | 77.6 | 74.7 | 75.5 | 76.3 | 78.2 |
| TIES-Merging | 78.3 | 76.5 | 75.8 | 75.8 | 74.7 | 77.1 |
| Diagonal Fisher merging | 79.8 | 76.2 | 79.8 | 75.1 | **79.8** | 75.1 |
| RegMean | – | 78.0 | – | 75.1 | – | **78.3** |
| MaTS | **83.0** | **79.4** | **80.5** | **76.9** | 78.7 | 77.3 |
| Sequential Training (UPPERBOUND) | 87.0 | 86.6 | 79.1 | 79.4 | 80.1 | 78.0 |
| Ensemble | 79.4 | 77.3 | 79.4 | 73.3 | 80.1 | 74.4 |

Table 2: RTE performance when merging a model fine-tuned on RTE with a model fine-tuned on various intermediate tasks. We **bold** the highest score.

Next, we consider intermediate-task model merging as introduced by Matena & Raffel (2022). Intermediate-task training (Phang et al., 2018; Pruksachatkun et al., 2020) involves first fine-tuning a model on an intermediate task before fine-tuning it on a target downstream task in hopes of improving performance. Instead of performing sequential training, intermediate-task model merging instead merges two models trained independently in parallel on the intermediate task and the target task. Following Matena & Raffel (2022), we consider fine-tuned BERT (Devlin et al., 2018) models with RTE (Dagan et al., 2005) as the target task

---

[3]We use the exact checkpoints released by Ilharco et al. (2022) at `https://github.com/mlfoundations/task_vectors`

and experiment with MNLI (Williams et al., 2017), QNLI (Rajpurkar et al., 2016; Wang et al., 2018), and QQP (Sharma et al., 2018) as the intermediate tasks. We compare the performance of `MaTS` with other merging methods, the original performance of the RTE-fine-tuned model, and the performance of actually doing sequential training in table 2. Overall, we find that `MaTS` often outperforms all baselines with the exception of intermediate-task merging with QQP, where most methods fail to provide a major improvement (perhaps because QQP is not a particularly effective intermediate task for RTE).

## 6.5 Cost

| Merging method | $(IA)^3$ | Full Model |
|---|---|---|
| Simple Averaging | 2.3E6 | 6.3E9 |
| Task Arithmetic | 4.8E6 | 1.3E10 |
| TIES-Merging | 5.0E7 | 1.8E11 |
| Diagonal Fisher merging | 1.2E10 | 3.2E13 |
| RegMean | – | 2.0E13 |
| MaTS | 1.3E14 | 1.2E14 |
| Multitask Training | 1.1E18 | 2.2E17 |

Table 3: FLOPS for various merging methods.

Having compared the performance of various merging methods, we now compare their computational costs. Specifically, we show the FLOPs required for `MaTS` (using the recipe found in section 6.2) and other merging methods in table 3. Though `MaTS` requires substantially more computation than other methods, most of the corresponding FLOPs consist of matrix multiplications which are efficient to perform on GPUs. Also, the number of FLOPs required for `MaTS` is still dramatically less than the FLOPS for multitask training. In practice, we found that merging with `MaTS` in the setting from section 6.2 only takes about 11 minutes on a single NVIDIA A6000 GPU, which is a relatively modest amount of time in the realm of training large neural networks. We provide the formulas used to derive the FLOP counts in appendix E.1 and provide wall-clock times of all methods in appendix E.2. We separately note that the storage costs required for the models and "metadata" required by each merging method is comparable and, for the models we consider, is small enough to fit on a single GPU.

## 6.6 Computing the Fisher

As discussed in section 2.2, there exist different perspectives as to whether it is appropriate to use the empirical Fisher (based on ground-truth labels) or true Fisher (based on the model's output distribution) in different settings. We therefore performed an ablation to compare using the empirical Fisher and the true Fisher in the experimental setting from section 6.2. For completeness, we also compare computing the Fisher on the training set or the validation set. Results are shown in table 4, where we confirm that the overall performance is best when using the empirical Fisher computed on the validation set.

| Merging method
Fine-tuning method | | Diagonal Fisher Merging
$(IA)^3$ | Full Model | Block-diagonal Fisher Merging
$(IA)^3$ | Full Model |
|---|---|---|---|---|---|
| Empirical Fisher | Train | 57.7 | 57.7 | 67.1 | 74.3 |
| | Validation | 59.1 | **63.8** | 68.8 | **76.1** |
| Fisher | Train | 59.6 | 56.6 | 67.1 | 74.2 |
| | Validation | **61.4** | 62.8 | **69.6** | 74.5 |

Table 4: Average multitask language model accuracy (following the same experimental setting as section 6.2) using the empirical or true Fisher for diagonal and block-diagonal Fisher merging.

## 6.7 Discussion

Overall, our experimental results confirm that `MaTS` significantly improves merging performance while incurring a manageable increase in computational costs. One notable finding is that using the block-diagonal Fisher merging (BFM) objective did not provide the best performance in the full-model fine-tuning setting despite the fact that we might expect it to provide the most reliable notion of a "task parameter subspace" among methods we consider (as discussed in section 5). We performed some additional analysis to explore

this discrepancy. First, we consider whether a better initialization could yield better performance with BFM. Specifically, since using a task vector-based initialization with the RegMean objective in `MaTS` performed well, we considered using the solution found by this recipe as an initialization for an additional round of `MaTS` using the BFM objective. In the experimental setting of section 6.2, adding an additional round of BFM-based merging increased the average performance across tasks from 72.5 to 73.5, outperforming all other methods. This result further emphasizes the importance of the choice of initialization in `MaTS` and also hints at the possibility of improved performance via multiple rounds of merging. Separately, we experimented with whether optimizing the BFM-based objective would benefit from more iterations of CG. However, when increasing the maximum number of iterations from 100 to 5K, we observed that the average performance could drop significantly and include graph in appendix G. We hypothesize this is due to numerical instability of the gradients. More broadly, however, our results generally suggest that improved approximations of the Fisher could improve the performance of Fisher merging-based methods.

# 7 Related Work

## 7.1 Loss Landscape

One common assumption for merging models is that the models lie in a "basin" of low loss in parameter space (Ilharco et al., 2022). There have been many past works focused on finding simple (not necessarily linear) paths of low loss between two models (Kuditipudi et al., 2019; Draxler et al., 2018; Garipov et al., 2018; Tatro et al., 2020; Benton et al., 2021). Entezari et al. (2021) hypothesize that all models trained with SGD can be permuted such that they are all linear mode connected (i.e. there is no barrier of high loss along the linear path connecting two models) (Frankle et al., 2020; Nagarajan & Kolter, 2019). The permutation symmetries present in neural networks imply that activations in models trained from different initializations may require permuting to put the models in the same loss basin. Neyshabur et al. (2020) show that models fine-tuned from the same pre-trained checkpoint lie in the same basin and Qin et al. (2022) studies the effect of hyperparameters on mode connectivity. Singh & Jaggi (2020); Ainsworth et al. (2022); Jordan et al. (2022); Peña et al. (2023) introduce various algorithms that account for soft or hard permutation symmetries in neural networks when performing merging and Stoica et al. (2023) extend this idea to merging models with different architectures. A related finding is the "monotonic linear interpolation property", which states that the loss of complex neural networks monotonically decreases when trained with SGD (Goodfellow et al., 2014) though models can be trained in such a way where this property does not always hold (Lucas et al., 2021). When linear mode connectivity does not hold, Lubana et al. (2023) show that models can use dissimilar mechanisms for making predictions and Juneja et al. (2022) show that non-mode-connected models generalize differently.

Past work has also aimed to analyze the structure of the loss landscape and try to directly learn a subspace of low loss. Some examples include Wortsman et al. (2021) learning a line, Hochreiter & Schmidhuber (1997) optimizing a box, or Fort & Jastrzebski (2019) finding an intersection of wedges. Other works have proposed that this "important subspace" can be found via the top eigenvectors of the Hessian (Fort & Ganguli, 2019), which correspond to the singular vectors of the covariance matrix (Izmailov et al., 2020), where the covariance can be estimated via SWAG (Maddox et al., 2019; Shwartz-Ziv et al., 2022) or other approximations such as the Laplace approximation (Maddox et al., 2020).

## 7.2 Fisher

There has been a great deal of work on better approximating the Fisher, usually in the context of using Natural Gradient Descent for optimization (Amari, 1998). K-FAC (Martens & Grosse, 2015) approximates the Fisher for linear layers as the Kronecker product of the covariance of the input activations and the covariance of the output activations. This have been extended to convolutional neural networks (Grosse & Martens, 2016) and recurrent neural networks (Martens et al., 2018). Other works have focused on reducing K-FAC's compute (Tang et al., 2021) or memory (Pauloski et al., 2021) and extending K-FAC for distributed training (Ba et al., 2017; Osawa et al., 2023). George et al. (2018) improve upon K-FAC by ensuring the

diagonal of the K-FAC approximation is unbiased. We note that using the Fisher is only one instance of the covariance matrix in our unified framework and `MaTS` uses the RegMean objective for full-model fine-tuning.

### 7.3 Additional Applications of Model Merging

Recently, many works have focused on merging models for different use cases. This includes merging models for intermediate task-training (Choshen et al., 2022; Gueta et al., 2023), generalizing better to out-of-distribution shifts (Wortsman et al., 2022b;a; Ramé et al., 2022), and combining modality-specific models into a multimodal model (Sung et al., 2023). Concurrently, other works have focused on inaccuracies in model merging stemming from gradient mismatches in different models, and use these insights to connect diagonal Fisher merging and Task Arithmetic Daheim et al. (2023).

## 8 Conclusion

We propose `MaTS`, a framework that connects past merging methods by viewing merging as matching individual models in their task parameter subspace. In general, using `MaTS` requires solving a linear system, so we propose using the conjugate gradient method which enables solving linear systems that would otherwise be intractable to solve. To explore a merging objective that has no tractable closed-form solution, we develop a variant of Fisher merging that makes a block-diagonal approximation of the Fisher. Experimentally, we show that `MaTS` achieves state-of-the-art results in merging to create multitask models from individual-task language and vision models and in intermediate-task merging. Our work provides a new perspective on model merging and motivates future work on improved methods for estimating a model's task parameter subspace.

### Acknowledgments

We thank Felix Dangel for pointing us to the conjugate gradient method for inverting a sum of Kronecker products, Juhan Bae for clarifying discussions on the Fisher, Michael Matena for clarifying discussions on the Fisher and valuable feedback on an earlier draft of this work, and Nikhil Kandpal for helpful discussions on how best to frame our work and valuable feedback on an earlier draft of this work.

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

# A Derivations

## A.1 Fisher Merging

Let $p$ be the number of parameters in the model and $\boldsymbol{\theta_i} \sim \mathcal{N}(\mu_i, \Sigma_i)$ The joint likelihood can be written as

$$\sum_{i=1}^{M} \log \left[ (2\pi)^{-\frac{p}{2}} \det \Sigma_i \exp \left( -\frac{1}{2}(\boldsymbol{\theta} - \boldsymbol{\theta_i}) \Sigma_i^{-1} (\boldsymbol{\theta} - \boldsymbol{\theta_i}) \right) \right]$$

The derivative of the joint log likelihood is:

$$\frac{\mathrm{d}}{\mathrm{d}\boldsymbol{\theta}} \sum_{i=1}^{M} \log P(\boldsymbol{\theta}|\boldsymbol{\theta_i}) = \frac{\mathrm{d}}{\mathrm{d}\boldsymbol{\theta}} \sum_{i=1}^{M} \log \left[ (2\pi)^{-\frac{p}{2}} \det \Sigma_i \exp \left( -\frac{1}{2}(\boldsymbol{\theta} - \boldsymbol{\theta_i}) \Sigma_i^{-1} (\boldsymbol{\theta} - \boldsymbol{\theta_i}) \right) \right]$$

$$= \frac{\mathrm{d}}{\mathrm{d}\boldsymbol{\theta}} \sum_{i=1}^{M} -\frac{p}{2} \log(2\pi) + \log \det \Sigma_i - \frac{1}{2}(\boldsymbol{\theta} - \boldsymbol{\theta_i}) \Sigma_i^{-1} (\boldsymbol{\theta} - \boldsymbol{\theta_i})$$

$$= \frac{\mathrm{d}}{\mathrm{d}\boldsymbol{\theta}} \sum_{i=1}^{M} -\frac{1}{2}(\boldsymbol{\theta} - \boldsymbol{\theta_i}) \Sigma_i^{-1} (\boldsymbol{\theta} - \boldsymbol{\theta_i})$$

$$= -\sum_{i=1}^{M} \Sigma_i^{-1}(\boldsymbol{\theta} - \boldsymbol{\theta_i})$$

$$= -\sum_{i=1}^{M} \Sigma_i^{-1}\boldsymbol{\theta} + \sum_{i=1}^{M} \Sigma_i^{-1}\boldsymbol{\theta_i}$$

Setting the derivative to 0 and solving, we get

$$\sum_{i=1}^{M} \Sigma_i^{-1}\boldsymbol{\theta} = \sum_{i=1}^{M} \Sigma_i^{-1}\boldsymbol{\theta_i}$$

$$\boldsymbol{\theta} = \left( \sum_{i=1}^{M} \Sigma_i^{-1} \right)^{-1} \left( \sum_{i=1}^{M} \Sigma_i^{-1}\boldsymbol{\theta_i} \right)$$

## A.2 RegMean

The minimization objective is below:

$$\min_{\boldsymbol{W}} \sum_{m=1}^{M} \frac{1}{N_m} ||\boldsymbol{O}_m - \boldsymbol{Z}_m \boldsymbol{W}||_2^2$$

$$= \min_{\boldsymbol{W}} \sum_{m=1}^{M} \frac{1}{N_m} (\boldsymbol{O}_m - \boldsymbol{Z}_m \boldsymbol{W})^{\top} (\boldsymbol{O}_m - \boldsymbol{Z}_m \boldsymbol{W})$$

$$= \min_{\boldsymbol{W}} \sum_{m=1}^{M} \frac{1}{N_m} \left( \boldsymbol{O}_m^{\top} \boldsymbol{O}_m - 2(\boldsymbol{Z}_m \boldsymbol{W})^{\top} \boldsymbol{O}_m + (\boldsymbol{Z}_m \boldsymbol{W})^{\top} (\boldsymbol{Z}_m \boldsymbol{W}) \right)$$

The gradient of the objective is

$$\sum_{m=1}^{M} \frac{1}{N_m} \left( -2\boldsymbol{Z}_m^{\top} \boldsymbol{O}_m + 2\boldsymbol{Z}_m^{\top} \boldsymbol{Z}_m \boldsymbol{W} \right)$$

Setting the derivative to 0 and solving, we get

$$\sum_{m=1}^{M} \frac{1}{N_m} 2\boldsymbol{Z}_m^\top \boldsymbol{O}_m = \sum_{m=1}^{M} \frac{1}{N_m} 2\boldsymbol{Z}_m^\top \boldsymbol{Z}_m \boldsymbol{W} \tag{15}$$

$$\sum_{m=1}^{M} \frac{1}{N_m} \boldsymbol{Z}_m^\top \boldsymbol{Z}_m \boldsymbol{W}_m = \sum_{m=1}^{M} \frac{1}{N_m} \boldsymbol{Z}_m^\top \boldsymbol{Z}_m \boldsymbol{W} \tag{16}$$

$$\boldsymbol{W} = \left( \sum_{m=1}^{M} \frac{1}{N_m} \boldsymbol{Z}_m^\top \boldsymbol{Z}_m \right)^{-1} \left( \sum_{m=1}^{M} \frac{1}{N_m} (\boldsymbol{Z}_m^\top \boldsymbol{Z}_m) \boldsymbol{W}_m \right) \tag{17}$$

### A.3   Task Parameter Subspace Derivation

$$\begin{aligned}
\boldsymbol{C}_m &= \boldsymbol{P}_m^\top \boldsymbol{P}_m \\
&= (\boldsymbol{U}_m \boldsymbol{\Sigma}_m \boldsymbol{V}_m^\top)^\top (\boldsymbol{U}_m \boldsymbol{\Sigma}_m \boldsymbol{V}_m^\top) \\
&= \boldsymbol{V}_m \boldsymbol{\Sigma}_m \boldsymbol{U}_m^\top \boldsymbol{U}_m \boldsymbol{\Sigma}_m \boldsymbol{V}_m^\top \\
&= \boldsymbol{Q}_m \boldsymbol{\Lambda}_m \boldsymbol{Q}_m^\top
\end{aligned}$$

### A.4   Connecting Fisher Merging and RegMean Derivation

$$\boldsymbol{F}_{ii}^l = \frac{1}{N} \sum_{n=1}^{N} \nabla_{\boldsymbol{W}_i} \log p_{\boldsymbol{\theta}}(y_n|x_n) \nabla_{\boldsymbol{W}_i} \log p_{\boldsymbol{\theta}}(y_n|x_n)^\top \tag{18}$$

$$= \frac{1}{N} \sum_{n=1}^{N} \boldsymbol{z}_n \boldsymbol{z}_n^\top \tag{19}$$

$$= \frac{1}{N} \boldsymbol{Z} \boldsymbol{Z}^\top \tag{20}$$

where $\boldsymbol{W}_i$ refers to the $i$th column of $W$.

### A.5   K-FAC Derivation

$$\begin{aligned}
\boldsymbol{W}^* &= \left( \sum_{m=1}^{M} \boldsymbol{F}_m \right)^{-1} \left( \sum_{m=1}^{M} \boldsymbol{F}_m \operatorname{vec}(\boldsymbol{W}_m) \right) \\
&= \left[ \sum_{m=1}^{M} \left( \frac{1}{N_m} \boldsymbol{Z}_m^\top \boldsymbol{Z}_m \right) \otimes \left( \frac{1}{N_m} \boldsymbol{O}_m'^\top \boldsymbol{O}_m' \right) \right]^{-1} \left[ \sum_{m=1}^{M} \left( \left( \frac{1}{N_m} \boldsymbol{Z}_m^\top \boldsymbol{Z}_m \right) \otimes \left( \frac{1}{N_m} \boldsymbol{O}_m'^\top \boldsymbol{O}_m' \right) \right) \operatorname{vec}(\boldsymbol{W}_m) \right] \\
&= \left( \sum_{m=1}^{M} \left( \frac{1}{N_m} \boldsymbol{Z}_m^\top \boldsymbol{Z}_m \right) \otimes \left( \frac{1}{N_m} \boldsymbol{O}_m'^\top \boldsymbol{O}_m' \right) \right]^{-1} \left[ \sum_{m=1}^{M} \left( \frac{1}{N_m} \boldsymbol{Z}_m^\top \boldsymbol{Z}_m \right) \boldsymbol{W}_m \left( \frac{1}{N_m} \boldsymbol{O}_m'^\top \boldsymbol{O}_m' \right) \right]
\end{aligned} \tag{21}$$

### A.6 Conjugate Gradient Derivation

$$
\begin{aligned}
\boldsymbol{b} &= \sum_{m=1}^{M} \boldsymbol{F}_m \operatorname{vec}(\boldsymbol{W}_m) \\
&= \sum_{m=1}^{M} \left[ \left( \frac{1}{N_m} \boldsymbol{Z}_t^{\top} \boldsymbol{Z}_m \right) \otimes \left( \frac{1}{N_m} \boldsymbol{O}_m'^{\top} \boldsymbol{O}_m' \right) \right] \operatorname{vec}(\boldsymbol{W}_m) \\
&= \sum_{m=1}^{M} \left( \frac{1}{N_m} \boldsymbol{Z}_m^{\top} \boldsymbol{Z}_m \right) \boldsymbol{W}_m \left( \frac{1}{N_m} \boldsymbol{O}_m'^{\top} \boldsymbol{O}_m' \right) \\
\boldsymbol{A}\boldsymbol{x} &= \left[ \sum_{m=1}^{M} \left( \frac{1}{N_m} \boldsymbol{Z}_m^{\top} \boldsymbol{Z}_m \right) \otimes \left( \frac{1}{N_m} \boldsymbol{O}_m'^{\top} \boldsymbol{O}_m' \right) \right] \operatorname{vec}(\boldsymbol{W}) \\
&= \sum_{m=1}^{M} \left[ \left( \frac{1}{N_m} \boldsymbol{Z}_m^{\top} \boldsymbol{Z}_m \right) \otimes \left( \frac{1}{N_m} \boldsymbol{O}_m'^{\top} \boldsymbol{O}_m' \right) \operatorname{vec}(\boldsymbol{W}) \right] \\
&= \sum_{m=1}^{M} \left( \frac{1}{N_m} \boldsymbol{Z}_m^{\top} \boldsymbol{Z}_m \right) \boldsymbol{W} \left( \frac{1}{N_m} \boldsymbol{O}_m'^{\top} \boldsymbol{O}_m' \right)
\end{aligned}
$$

## B   Conjugate Gradient Method

Instead of using the gradient of the linear objective $\boldsymbol{A}\boldsymbol{x} - \boldsymbol{b}$ though, CG uses the gradient of a convex quadractic objective $\frac{1}{2}\boldsymbol{x}^{\top}\boldsymbol{A}\boldsymbol{x} - \boldsymbol{x}^{\top}\boldsymbol{b}$. Both the linear objective and the convex quadratic objective have the same minimum. The advantage of using the quadractic objective to compute the gradient is that its gradient is equal to $\boldsymbol{A}\boldsymbol{x} - \boldsymbol{b}$, which is just the "error" of our current estimate of $x$ in the linear objective. CG additionally constrains all update directions computed at every iteration to be conjugates. While two vectors are consider orthogonal if their dot product is 0, two vectors are conjugates if their dot product with respect to a matrix (in this case $\boldsymbol{A}$) is 0. Constraining the update directions to be conjugates implies that all future updates do not affect the optimality of a previous update, and thus the optimal step size for a given update direction can be computed. This ultimately ensures that the number of iterations required for convergence is upper bounded by the dimension of the linear system. In practice, we observe the number of iterations necessary for convergence to be much smaller since the update direction corresponds to the gradient of an objective with the same minimum.

## C   Algorithm

We show the algorithm for `MaTS` in the $(IA)^3$ setting in algorithm 1. In the full-model setting, the outer product of the gradients is replaced with the outer product of the input activations.

---

**Algorithm 1:** MaTS Algorithm

---

**Input** : $\theta_1...\theta_M$            $\triangleright$ $M$ fine-tuned models sharing a common architecture and initialization

**Input** : $D_1 \ ... \ D_M$            $\triangleright$ $M$ validation datasets to compute model statistics

**Output:** $\theta^*$            $\triangleright$ Merged model

```
// For each task, compute the Fisher.  For brevity, we only show computing the Fisher
   for each layer.
```

**for** $m = 1, \ m \le M, \ m + 1$ **do**

     $F_m = 0$

     **for** $(x, y) \in D_m$ **do**

         |   $F_m \mathrel{+}= \nabla_{\boldsymbol{\theta}} \log p_{\boldsymbol{\theta_m}}(y|x) \nabla_{\boldsymbol{\theta}} \log p_{\boldsymbol{\theta_m}}(y|x)^\top$            $\triangleright$ Outer product of per-example gradient

     **end**

     $F_m = F_m/|D_m|$            $\triangleright$ Normalize Fisher by number of examples

**end**

$b = \sum\limits_{m=1}^{M} F_m \theta_m$            $\triangleright$ $b$ in the linear system $Ax = b$

$\theta = \theta_{\texttt{task\_arithmetic}}$            $\triangleright$ Initialize $\theta$ using Task Arithmetic

```
// Conjugate Gradient Method
```

**for** $i = 1, \ m \le$ *Number of Iterations*, $i + 1$ **do**

     `Update` $\theta$ `using conjugate gradient where`   $Ax = \sum\limits_{m=1}^{M} F_m \theta$

**end**

**return** $\theta$

---

# D  Experimental Setup

## D.1  Fine-tuning Details

For all the models we fine-tuned, we use the same hyperparameter setup to be consistent. Concretely, we use AdamW, learning rate of $1e^{-4}$, batch size of 1024, bfloat16 during training, training for 5K batches, checkpointing every 100 batches, and early stopping if the model has not improved for 5 checkpoints. For $(IA)^3$, the only difference is the learning rate is set to $5e^{-3}$.

We use the templates from PromptSource (Bach et al., 2022) to format the input/output pairs in natural language. All the models were trained and evaluated on all the templates available in PromptSource. Instead of evaluating the dataset across all templates, we randomly assign a fixed template to each datapoint and evaluate the dataset once. This is equivalent to averaging the performance across templates.

## D.2  8 Keys tasks for the T0 Mixture

The tasks and datasets were question-answering (CosmosQA (Huang et al., 2019), SocialIQA (Sap et al., 2019), QuAIL (Rogers et al., 2020), Wiki QA (Cohen et al., 2018), QuaRTz (Tafjord et al., 2019), QASC (Khot et al., 2020), ROPES (Lin et al., 2019)) and paraphrasing (PAWS (Zhang et al., 2019)).

## D.3  TIES Mixture

The datasets are PAWS (Zhang et al., 2019), QASC (Khot et al., 2020), QuaRTz (Tafjord et al., 2019),Story Cloze (Sharma et al., 2018), Wiki QA (Cohen et al., 2018), Winogrande (Sakaguchi et al., 2020), and WSC (Levesque et al., 2012).

## D.4  Vision Tasks

These checkpoints were initialized from CLIP (Radford et al., 2021) and individually fine-tuned on eight image classification datasets: Cars (Krause et al., 2013), DTD (Cimpoi et al., 2014), EuroSAT (Helber

et al., 2019; 2018), GTSRB (Houben et al., 2013), MNIST (Yann, 1998), RESISC45 (Cheng et al., 2017), SUN397(Xiao et al., 2016), and SVHN (Yuval, 2011).

## E    Cost

### E.1    FLOPS Derivation

We assume each model has $p$ parameters, or $|\boldsymbol{\theta}| = p$ and the total number of tokens and examples used to compute the model statistics (i.e. the Fisher or the covariance of the input activations) are $T$ and $N$ respectively. We follow Kaplan et al. (2020); Liu et al. (2022) to estimate the FLOPS to compute the gradient (forward pass + backward pass) as $3pT$ and the number of FLOPS to compute a forward pass in a model is $pT$.

Averaging requires $Mp$ FLOPS. Summing the models requires $(M-1)p$ FLOPS and dividing by the total number of models requires another $p$ FLOPS.

Task Arithmetic requires $2Mp + p$ FLOPS. Computing the task vector requires $Mp$ FLOPS, adding task vectors requires $(M-1)p$ FLOPS, adding the pretrained model requires $p$ FLOPS, and multiplying the final task vector by the scalar $\lambda$ requires another $p$ FLOPS.

Diagonal Fisher merging requires $3Mp - p + M(3pT + Np)$ FLOPS. Computing the gradient requires $3pt$ FLOPS. Computing the diagonal Fisher requires $Np$ FLOPS, since it is an element-wise multiplication for each gradient computed per-example. This results in $M(3pT + Np)$ FLOPS to compute the diagonal Fisher across all model. Scaling each model by its diagonal Fisher requires $Mp$ FLOPS, adding the models scaled by the diagonal Fisher requires $(M-1)p$ FLOPS, adding the diagonal Fishers requires $(M-1)p$ FLOPS, and dividing the sum of the scaled models by the sum of the diagonal Fishers requires another $p$ FLOPS.

TIES requires $9.8Mp + Mp \log p - 2p$ FLOPS. The trimming stage requires a total of $1.8Mp + Mp \log p$ FLOPS. $Mp$ FLOPS to compute the magnitude for each parameter across all models, $Mp \log p$ FLOPS to sort the magnitudes for each model, and $0.8Mp$ FLOPS to reset the bottom the 0.8 parameters by magnitude to 0. The elect stage requires a total of $6Mp - p$ total FLOPS. For the elect stage, computing the positive mass requires $Mp$ FLOPS to compute the sign for each parameter across all models, $Mp$ FLOPS to multiply a parameter by its sign, $(M-1)p$ FLOPS to sum the positive mass for models. Computing the negative mass doubles the amount of FLOPS. Comparing the positive and negative mass and keeping the larger mass requires another $p$ FLOPS. The disjoint merging stage requires a total of $2Mp - p$ FLOPS. This includes $MP$ FLOPS to multiply the parameters by the correct sign (or 0 if the parameter should be masked out) and another $(M-1)p$ FLOPS to sum the parameters.

RegMean requires $Mpt + \sum_{l=1}^{L} (M-1)d_l^2 + \frac{2}{3}d_l^3 + Md_l^2 k_l + (M-1)d_l k_l + d_l^2 k_l + N d_l^2$ FLOPS, where $d_l$ and $k_l$ are the input dimension and output dimension for layer $l$. For a given linear layer, summing the gram matrices requires $(M-1)d^2$ FLOPS, inverting the sum of the gram matrices requires $\frac{2}{3}d^3$ FLOPS, multiplying the gram matrices by the model weight requires $Md^2 k$ FLOPS, summing the model weights modified by the gram matrices requires $(M-1)dk$ FLOPS, and multiplying the final 2 terms require $d^2 k$ FLOPS. Computing the forward pass activations requires $pT$ FLOPS per model.

We assume MaTS requires $I$ iterations. For merging full models, MaTS requires $MpT + \sum_{l=1}^{L} (M-1)d_l^2 + Md_l^2 k_l + (M-1)d_l k_l + I(d_l^2 k_l + 12 d_l k_l) + T d_l^2$ FLOPS. For merging $(IA)^3$, MaTS requires $3MpT + \sum_{l=1}^{L} (M-1)d_l^2 + Md_l^2 k_l + (M-1)d_l k_l + I(d_l^2 k_l + 12 d_l k_l) + N d_l^2$ FLOPS

For a given layer, summing the gram matrices and summing the models multiplied by the gram matrices requires $(M-1)d^2$ and $Md^2 k + (M-1)dk$ FLOPS respectively just as in RegMean. Within each iteration, the matrix vector product requires $d^2 k$ FLOPS and the 12 vector operations within each iteration to compute

the step size require $12dk$ FLOPS. This means $I$ iterations requires $I(dk + 12d^2k)$ FLOPS. Computing the input activations covariances for linear layers requires $Td^2$ FLOPS, since it is a matrix multiplication between a matrix of size $d \ x \ T$ and $T \ x \ d$ and the blockwise Fisher matrices for $IA3$ requires $Nd^2$ FLOPS since it is a matrix multiplication between a matrix of size $d \ x \ N$ and $N \ x \ d$. Computing the forward pass activations requires $pT$ FLOPS per model and computing the gradient requires $3pT$ FLOPS per model.

For multitask learning, we follow Kaplan et al. (2020); Liu et al. (2022) to estimate the FLOPS to train an encoder-decoder model as $3pT$ where $T$ is the total number of tokens during training. The batch size is 1024 and the average sequence length is 138.5. For full model fine-tuning, the model was trained $2K$ batches and for $(IA)^3$, the model was trained $10K$ batches.

For full-model fine-tuning, there are 783M parameters, with 288 linear layers that have input and output dimension 1024, 96 layers that have input dimension 1024 and output dimension 2816 and 48 linear layers that have input dimension 2816 and output dimension 38. For $(IA)^3$, there are 282K parameters, with 144 vectors having dimension 1024 and 48 vectors having dimension 2816. The number of tokens used to estimate the Fisher is roughly $T = 1385$ since the average sequence length is 138.5 and the Fisher is computed on $1K$ examples.

### E.2 Wall Clock Time

In table 5, we show the wall-clock time for merging various methods on an A6000 machine. The wall-clock time only includes computing the merge (and statistics for methods that require them) and not actually loading any checkpoints. Though MaTS is slower than other methods, it is only by a factor of $2 - 3x$ for full-model fine-tuning and is less than 6 seconds for $(IA)^3$.

| Method | Time (seconds) | |
| --- | --- | --- |
| | $(IA)^3$ | Full Model |
| Avg. | $1.1E^{-2}_{9.5E-3}$ | $6.0_{2.0}$ |
| Task Arithmetic (Ilharco et al., 2022) | $1.7E^{-2}_{2.4E-3}$ | $8.9_{1.7}$ |
| TIES (Yadav et al., 2023) | $4.0E^{-2}_{5.4E-3}$ | $75.7_{11.9}$ |
| DFM (Matena & Raffel, 2022) | $1104_9$ | $1272_{15}$ |
| RM (Jin et al., 2022) | $-$ | $665_{17.7}$ |
| MaTS | $1128_{3.1}$ | $1287_{28}$ |

Table 5: Wall-clock time for various merging methods. We report the mean and standard deviation across 10 runs.

## F  Full Results

The full results of the performance of various merging methods on the individual tasks is shown below. This includes full-model fine-tuning and $(IA)^3$ on the 8 key tasks from the T0 training mixture and the 7 tasks from the TIES mixture on T5-Large.

## G  Performance vs. Number of Iterations

We include a graph of the performance of the merged model and the error in solving the linear system vs. the number of iterations of the conjugate gradient method in fig. 3. We show the performance using the block-diagonal Fisher merging objective on $(IA)^3$ and full-model fine-tuning and the RegMean Jin et al. (2022) objective on full-model fine-tuning.

In most cases, we do find that the merging objective value decreases as the number of iterations increases. In some cases, we observe a small amount of noise in the merging objective, which we attribute to numerical instability, but even in these cases the objective value tends to decrease. In some cases, we do not observe the performance (e.g. accuracy) to increase as the number of CG iterations increases.

| Method | Avg. | CosmosQA | SocialIQA | PAWS | QuAIL | WikiQA | QuaRTz | QASC | ROPES |
|---|---|---|---|---|---|---|---|---|---|
| Avg. | 56.4 | 41.8 | 51.6 | 55.7 | 49.3 | 66.3 | 62.2 | 82.6 | 41.4 |
| Task Arithmetic | 63.8 | 54.5 | 56.9 | 61.1 | 56.9 | 76.5 | 76.6 | 79.0 | 48.7 |
| TIES | 62.8 | 55 | 60.1 | 88.7 | 58.5 | 35.9 | 80.2 | 76.6 | 47.3 |
| DFM | 57.7 | 44.9 | 51 | 51.5 | 56.1 | 49.5 | 81.5 | 78.0 | 49.4 |
| RegMean | 69.1 | 47.6 | 59.1 | 79.6 | 54.2 | 88.9 | 80.5 | 94.2 | 48.3 |
| MaTS | 72.5 | 53.7 | 60.2 | 84.2 | 56.5 | 94.0 | 85.2 | 94.8 | 51.6 |
| Ensemble | 64.9 | 50.0 | 51.1 | 89.2 | 56.1 | 72.6 | 78.4 | 79.2 | 42.3 |
| Multitask | 80.5 | 69.3 | 62.3 | 94.7 | 64.9 | 95.4 | 89.1 | 97.8 | 70.4 |

Table 6: Accuracy of different merging methods on the 8 keys tasks from the T0 training mixture when using full-model fine-tuning.

| Method | Avg. | CosmosQA | SocialIQA | PAWS | QuAIL | WikiQA | QuaRTz | QASC | ROPES |
|---|---|---|---|---|---|---|---|---|---|
| Avg. | 48.1 | 37.4 | 47.6 | 48.6 | 43.3 | 37.1 | 58.9 | 81.6 | 30.7 |
| Task Arithmetic | 57.6 | 43.9 | 53.0 | 59.0 | 52.3 | 49.4 | 74.0 | 88.4 | 41.1 |
| TIES | 56.8 | 40.7 | 50.4 | 76.1 | 47.9 | 52.2 | 72.9 | 75.0 | 39.3 |
| DFM | 55.9 | 40.1 | 46.4 | 52.0 | 46.2 | 44.5 | 87.5 | 77.0 | 53.9 |
| MaTS | 65.1 | 48.5 | 53.2 | 65.8 | 55.2 | 77.8 | 77.6 | 87.8 | 55.0 |
| Ensemble | 62.8 | 49.4 | 50.6 | 79.5 | 51.3 | 61.7 | 84.9 | 88.2 | 36.8 |
| Multitask | 76.7 | 64.3 | 59.6 | 94.5 | 60.7 | 95.5 | 83.9 | 96.4 | 58.5 |

Table 7: Accuracy of different merging methods on the 8 keys tasks from the T0 training mixture when using $(IA)^3$.

| Method | Avg. | PAWS | QASC | QuaRTz | StoryCloze | WikiQA | Winogrande | WSC |
|---|---|---|---|---|---|---|---|---|
| Avg. | 60.5 | 55.9 | 71.6 | 54.4 | 56.4 | 76.7 | 53.4 | 54.8 |
| Task Arithmetic | 71.9 | 76.5 | 64.2 | 70.6 | 80.5 | 90.9 | 65.5 | 54.8 |
| TIES | 69.6 | 89.5 | 78.8 | 78.1 | 73.8 | 36.0 | 73.2 | 57.7 |
| DFM | 64.0 | 55.7 | 75.6 | 85.9 | 69.6 | 50.4 | 59.0 | 51.9 |
| RegMean | 78.9 | 82.2 | 94.8 | 76.6 | 80.6 | 93.9 | 61.6 | 62.5 |
| MaTS | 81.5 | 86.6 | 95.2 | 81.8 | 83.0 | 94.5 | 66.6 | 62.5 |
| Ensemble | 74.1 | 88.9 | 60.6 | 76.3 | 74.6 | 92.2 | 64.7 | 61.5 |
| Multitask | 84.5 | 95.2 | 97.2 | 87.5 | 92.1 | 95.9 | 74.5 | 49.0 |

Table 8: Accuracy of different merging methods on the tasks from the TIES training mixture for T5-Large when using full-model fine-tuning.

| Method | Avg. | PAWS | QASC | QuaRTz | StoryCloze | WikiQA | Winogrande | WSC |
|---|---|---|---|---|---|---|---|---|
| Avg. | 52.1 | 49.7 | 67.8 | 53.4 | 57.4 | 39.0 | 52.5 | 45.2 |
| Task Arithmetic | 67.1 | 55.2 | 75.0 | 69.5 | 77.7 | 75.8 | 58.9 | 57.7 |
| TIES | 66.0 | 70.1 | 80.6 | 72.9 | 73.3 | 56.2 | 58.1 | 51.0 |
| DFM | 61.4 | 51.2 | 73.0 | 89.3 | 72.8 | 45.0 | 57.1 | 41.3 |
| MaTS | 75.3 | 75.6 | 83.8 | 78.4 | 81.6 | 86.6 | 61.6 | 59.6 |
| Ensemble | 74.8 | 85.4 | 79.6 | 83.6 | 87.8 | 72.8 | 60.4 | 53.8 |
| Multitask | 82.2 | 94.9 | 95.8 | 81.2 | 88.2 | 95.3 | 66.4 | 53.8 |

Table 9: Accuracy of different merging methods on the tasks from the TIES training mixture for T5-Large when using $(IA)^3$.

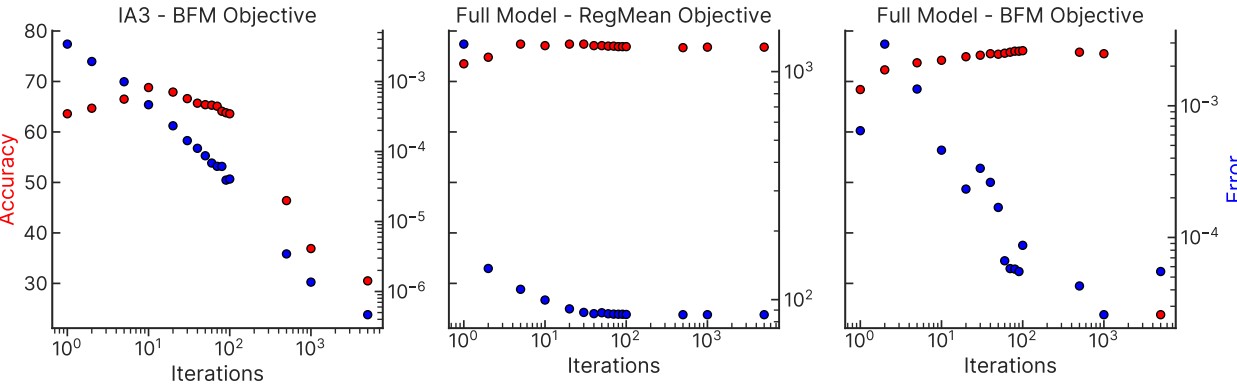

Figure 3: Performance of the merged model and the in error solving the linear system vs. the number of iterations in the conjugate gradient method.

