# OpenReview forum: "Merging by Matching Models in Task Parameter Subspaces"
_TMLR — Accepted by TMLR_

### Review · Reviewer_pzCj · 2024-01-19

**Summary Of Contributions:**

The paper casts few neural network parameter averaging methods as unified covariance-averaging, and presents an efficient conjugate-gradient method to find the optimal parameter averagings.

**Audience:**

Yes

**Broader Impact Concerns:**

-

**Claims And Evidence:**

No

**Requested Changes:**

This paper studies how to merge multiple neural networks into one combined one by averaging the parameters in different ways. The paper considers three merging approaches (average, Fisher average, activation matching), and casts them through unified covariance-weighted averaging, and taking its SVD form. The paper presents a conjugate-gradient method to compute the mergings in practise. The paper seems to have two novel contributions: unified merging math, and the CG.

The methods in this paper are all sensible and principled, but quite textbook. The paper is written in a clear and intuitive style.

The paper frames its discussion against “task subspaces”. I do not agree with this. I don’t think there are any “tasks” here, nor do I think the SVD represents any tasks. The paper simply merges different DNN weights, and this does not need to have anything to do with tasks. The SVD represents the parameter subspaces around different modes of the same posterior, or across slightly varying posteriors. These posteriors can come from simply re-trainings, or from having different datasets, but they can’t come from having different tasks since then the parameter vectors would not be same size (which they need to be). I guess one could have multiple eg. classification tasks as multiple single-task networks, which are merged, but this makes little sense. Can you discuss this?

The paper does not acknowledge that the SVD inverse simplifies into one where the inverse is only over eigenvalues. The paper motivates their CG optimiser due to not being able to do the inverse, so I wonder if the motivation for CG dissolves?

The paper needs to include simple ensemble as a baseline, ie. using $f(x) = 1/M \sum_m f_m(x)$, or something similar. The paper also needs to include the individual models $\theta_m$ as baselines as well.

I don’t think Fig1 makes much sense. The markers are placed all over the place, and I can’t see any rhyme or reason why the stars/diamonds are where they are, or what the lines mean. The figure also doesn’t even show any mergings: where are the averaged points? This figure needs to be reworked.

I did not understand what the MATS actually is. The paper talks about various merging things, and CG and blocks, etc. What is MATS?

I did not understand what the paper means by tasks. The parameter merging clearly requires the parameters to be of same size, and have same meaning. Hence, I think the merging only applies to the “body” of the network, and one can’t merge the different heads, where one would expect each task to have one head. Yet, the text always talks about tasks, even though we clearly can’t merge anything task-specific. Err… so what’s going on?

To clarify my stance: I’m thinking here of eg. BERT/CLIP style foundation models where we have a massive embedding network that produces high-quality embeddings of the inputs (=body), and then we have smaller networks that map these embeddings into different predictions needed by different tasks (=heads). We want to train the heads; and either keep the body frozen or perhaps only slightly re-optimise it for each head. Can you clarify what kind of setting you are thinking in this paper? Are you merging the body or the heads, or something else?

I’m also confused what are the tasks in the first place. Are we doing classification? Do we have multiple classes or just one many times? The experiments bring in some kind of task vectors, without any math, or any explicit connection to the earlier math. What are these and how do they go into the method?

Are different models re-trainings, or different datasets, or different classes, or different labellings of same classes, etc..? I also don’t understand what the different models theta_m are or where do they come from.

Also, why do we want to merge models in the first place? If we have 8 fine-tuned CLIP models, surely we want to retain those and not mess up them into one hybrid model.

If the multitask seems to always be superior, why do we want to merge? Why not use the multitask model instead?

I did not understand 6.4. So we have an RTE-optimised method, which gets 79.8, and then we have intermediate-optimised method that is combined to it to get 83.0. Ok, but where is the intermediate-optimised method itself? Surely you want to have one method for RTE-opt, one for int-opt→RTE-opt, and finally a merged one. There also seems to be all kinds of finetunings going on. It’s difficult to see in the results what values are caused by what, since lots of stuff is happening behind the curtains.

Minor comments

- The paper mixes $m$ and $\bf{m}$ and $\bf{\theta}$ and $\theta$ randomly, which makes the manuscript look unfinished. Please have all math consistent wrt boldfacing and upper/lowercases.
- The paper implicitly assumes that all models have same parameters with same meaning: otherwise eq 1 wouldn’t make any sense. This should be made explicit: this paper considers only merging different optimisations of the same network. This limits the paper’s scope.
- Eq 1 should cite and discuss SWA(G) (Wilson et al)
- eq (unnumbered, after eq 2 [please number all equations]) has F while it should have F_m.
- What is $1/\sum_m F_m$? Is this an elementwise inverse, or matrix inverse? Is this before or after the diagonal approximation? Later these inverses seem to be matrix ones, but here it looks like elementwise. This is likely a mistake. Please clarify.
- It’s a bit strange that eq 5 does not acknowledge the obvious simplification $(QLQ^T)^{-1}$ = $QL^{-1}Q^T$. Surely you want to do this?
- Please clarify what you mean by finetuning. It is an ambiguous term.
- For vision, what was the CLIP trained against? What did you finetune? Why can’t you just train all 8 label sets together? Surely it’s way more expensive to train 8 tasks separately, than just one combined. What is the test setting here? Do you mix all images from all 8 tasks? How do you handle resolutions? How many classes are there in the 8? I’m quite confused what the experimental setting here is.
- What does “base model” mean in the introduction? I’m a bit confused of the motivation of this work. Let’s take the Stable Diffusion XL example. There are surely some specialised models that eg. only draw cartoon figures and nothing else. There are also surely some models that specialise in some artistic style. If you average these models, you surely just get gibberish. What are you then trying to achieve? Surely the specialised models need to have considerable overlap in their domains. Can you clarify your setting?
- I struggled to understand the experiments. Could you explain them in higher level: what are you trying to achieve? What kind of models do you want to merge, and why? Illustrations or math would be nice (eg. what are the sizes, dimensions, outputs).
- What are the assumptions for your models to be merged?

**Strengths And Weaknesses:**

The methods are sensible and principled.

The topic is interesting and important.

The results are promising and impressive at times.

The paper writing has major issues, and I did not really understand this paper.

---

> ### Author Response · Authors · 2024-02-15
> **Response to Reviewer pzCj**
>
> Thank you for your detailed review and constructive suggestions. Your perspective has been particularly helpful in uncovering where our paper assumes familiarity with the problem setting and lacks useful background information, and we have updated our draft accordingly.
>
> First, in multiple places your review raises the question of what we mean by a "task", what assumptions we make when merging models, and why one would want to merge models in the first place. While our work follows a problem setting and definitions that are widespread in a large body of past work, we clarify this background below and add it to our revised submission:
>
> **What is a task?** A "task" is simply the input-output relationship that we aim to train a model to perform. For example, "sentiment analysis" is a common task in natural language processing where the task is to identify the sentiment (positive, neutral, negative, etc.) of a piece of text. A task does not necessarily dictate a specific choice of architecture, loss function, or task-specific parameters; for example, sentiment analysis could be performed via regression (outputting a continuous value representing positive vs. negative), classification, or even text generation (literally outputting "This text is positive", etc.). This definition of "task" is widespread in the literature. [3, 6, 7, 8, 9, 10, 11, 12, 13, 14, 15, 16, 17, 18]
>
> **What is model merging and why would we want to do it?** When we merge models, we aim to construct a single model that retains the capabilities of the individual models being merged. A common example application of merging is constructing a multitask model from individual-task models, which is the primary application we explore in our paper. Compared to multitask learning, merging does not require simultaneous access to the individual-task datasets. Compared to output-space ensembling of $M$ models, merging produces a model that is $M$ times cheaper to run inference with. In addition, merging provides a path to recycle preexisting individual-task models that are widely shared on model hubs to flexibly and cheaply create new models. Our work contributes to a large body of literature on merging that uses the same definition and motivations. [1, 2, 4, 19, 20, 21, 22, 23, 24, 25, 26, 27, 28, 29, 30, 31, 32, 33, 34]
>
> **What assumptions does merging make?** Our work, as in almost all past work on model merging [1, 2, 4, 34], assumes that the individual models are all fine-tuned variants of a single base model. By "fine-tuned", we mean that a base pre-trained model has undergone additional training on a task-specific dataset. The individual models therefore share an architecture and have a natural mapping between their parameters, so that e.g. simply averaging together the parameters of the individual models is feasible. For "open vocabulary" models like T5 [3] (used in our NLP experiments), there are no task-specific parameters or "classification heads" in the fine-tuned models because all tasks use a text-to-text format. For models like CLIP [7] (used in our vision experiments) that introduce a classification head on top of a "body" when fine-tuning on a target task, only the "body" parameters are merged while the "head" parameters are kept intact from the individual models. This matches standard practice when merging models that have separate task heads [1, 2, 4, 34].
>
> **What is a "task subspace"?** In the introduction, our paper defines a task subspace as "the subspace implicitly used by a given merging method that aims to correspond to the important dimensions in parameter space for the task", which we later precisely relate to $\mathbf{Q}_m$. Intuitively, the task subspace denotes the dimensions in parameter space that are important for a particular task (i.e., changing their value has a high influence on performance on the task). While past works have made use of similar notions in model merging [2, 34], we are the first to formalize this notion, relate the different works, and build a framework around it.
>
> We responded below in-line to the questions and would be glad to clarify anything else.

---

> > ### Comment · Reviewer_pzCj · 2024-02-19
> > **resp**
> >
> > Thanks for clarifications. I still have comments.
> >
> > - In your response you state that your merging requires a common pre-trained base model, and a bunch of models that each come with their own head, and some fine-tuning of the base parameters. This is a specific setting, and the paper doesn’t make these assumptions explicit. In contrast, sec 2.0 claims that one simply has a bunch of networks with common architecture and a single y. There is no discussion of bases or heads or tasks. The paper should introduce different tasks notationally, and also split the parameter vector into base/head parts, and make it explicit in the notation that only the base parts are merged (or are they..?), and operated over in the rest of the paper.
> >
> > - Also, what happens to heads in merging?
> >
> > - It would also be good to define what you mean by “fine-tuning”, since its an overloaded term.
> >
> > - I still don’t understand the task subspace idea. There is some vague hand-waving about this in sec 3.0., which I can’t follow. Sec 3.2. has some more vague'ish discussion of this concept. I can't find a clear explanation of this idea, nor its motivation, significance or introduction. The SVD finds a parameter subspaces, and I fail to see how this somehow represents the tasks. Sure, different model SVD’s correspond to the parameter bases of different models, but why do we care about this? I don’t see this notion going anywhere or contributing anything. What are you trying to achieve or convey by talking about “task subspaces”?
> >
> > - Furthermore, the eigenvectors of different model parameters all point in different directions, and are not comparable. I think this would make sense if there is some shared latent space where merging happens, but this does not seem to be true. I also don’t see the “task subspaces” as actually doing anything: the improvements to merging come solely from the Fisher or RegMean covariances, and the SVD doesn’t change anything. So… what’s then the point of any of this?
> >
> > - Finally, in sec 5 you don’t talk about Q or Lambda at all anymore, and instead everything is done using Z’s. If we don't even use the Q_m or L_m, why did we talk about them?
> >
> > - Or maybe you are trying to talk about “task’s parameter subspaces” instead of “task subspaces”? That would make sense: each task has a different parameter decomposition. But even this is a big vague: this is really the task's loss function's parameters subspace. A “task subspace” to me refers to an embedding of the task itself (ie. SVD of y). Please define more rigorously what you mean by “task subspace”.
> >
> > - After eq 8 paper claims that F_m is a per-example covariance. I assume example means a single datapoint (x,y), but Fisher computes over entire dataset D_m. Maybe “example” refers to “task”?  Please clarify.
> >
> > - The Qm in Fisher case is a first-order approximation of the second-order Hessian, which is very inaccurate, especially when we talk about million/billion dimensional matrices. Why does it make sense to claim that Qm are viewed as Hessian eigenvectors: surely these are very different things?
> >
> > - The discussion in sec 3.2. states that a projection QLQ will upweight important components. This is misleading. The decomposition will do this only if the underlying covariance C_m does this, and the decomposition itself doesn’t really do anything here. Hence, you should instead discuss here the specific effect of specific Cm choices, and not claim that the SVD does parameter relevance detection.
> >
> > - Please include an algorithm block of the overall method.

---

> > > ### Author Response · Authors · 2024-02-22
> > > **Response**
> > >
> > > Thank you for your feedback and questions. We respond in-line to your questions below.
> > >
> > >
> > > >  In your response you state that your merging requires a common pre-trained base model, and a bunch of models that each come with their own head, and some fine-tuning of the base parameters. This is a specific setting, and the paper doesn’t make these assumptions explicit. In contrast, sec 2.0 claims that one simply has a bunch of networks with common architecture and a single y. There is no discussion of bases or heads or tasks. The paper should introduce different tasks notationally, and also split the parameter vector into base/head parts, and make it explicit in the notation that only the base parts are merged (or are they..?), and operated over in the rest of the paper.
> > >
> > > Merging is performed on fine-tuned variants of a common pre-trained base model. Sec 2.0 states that the given models “all share a common architecture and initialization”. Merging does not require or assume that the models each have their own head. For our experiments in NLP, we use the widespread text-to-text setting where the models do not come with their own bases/heads/tasks and **every** parameter is merged. In the vision setup, which we follow from [1] for a fair comparison, the models come with their own heads, but this is not an assumption or requirement of our method (or any merging method).
> > >
> > > > Also, what happens to heads in merging?
> > >
> > > For the vision setup only, following [1] for a fair comparison, the individual heads are not merged and the head corresponding to the target task is used for inference on a particular task.
> > >
> > > > It would also be good to define what you mean by “fine-tuning”, since its an overloaded term.
> > >
> > > By fine-tuning, we mean that a pre-trained model (i.e. a model whose parameters are not just randomly initialized but are the result of training on some data) is further trained on a task- or domain-specific dataset. If you believe we should disambiguate from other uses of the term that are common, please let us know.
> > >
> > > > I still don’t understand the task subspace idea. There is some vague hand-waving about this in sec 3.0., which I can’t follow. Sec 3.2. has some more vague'ish discussion of this concept. I can't find a clear explanation of this idea, nor its motivation, significance or introduction. The SVD finds a parameter subspaces, and I fail to see how this somehow represents the tasks. Sure, different model SVD’s correspond to the parameter bases of different models, but why do we care about this? I don’t see this notion going anywhere or contributing anything. What are you trying to achieve or convey by talking about “task subspaces”?
> > >
> > > > Or maybe you are trying to talk about “task’s parameter subspaces” instead of “task subspaces”? That would make sense: each task has a different parameter decomposition. But even this is a big vague: this is really the task's loss function's parameters subspace. A “task subspace” to me refers to an embedding of the task itself (ie. SVD of y). Please define more rigorously what you mean by “task subspace”.
> > >
> > >  The task subspace gives a way to interpret eq. (4). Specifically, the task subspace consists of the basis vectors $Q_m$ whose relative importance is defined by $\Lambda_m$. Since each model is assumed to be fine-tuned on a different task, we can therefore interpret $Q_m$ and $\Lambda_m$ respectively as the relative importance of different basis vectors for some particular task. To us, this motivates the term "task subspace", but we would be open to using "task's parameter subspaces" if you think it would be clearer or more precise.
> > >
> > > > Furthermore, the eigenvectors of different model parameters all point in different directions, and are not comparable. I think this would make sense if there is some shared latent space where merging happens, but this does not seem to be true. I also don’t see the “task subspaces” as actually doing anything: the improvements to merging come solely from the Fisher or RegMean covariances, and the SVD doesn’t change anything. So… what’s then the point of any of this?
> > >
> > > The eigenvectors of different models point in different directions, but since these models share a common architecture, these directions in parameter space have a common meaning. The task subspace is not modifying the Fisher or Regmean covariances (and so there will be no change in performance). It gives a unified interpretation of the Fisher and RegMean covariances, which allows us to cast various merging methods as optimizing some particular objective. We show that this objective can be straightforwardly optimized with the conjugate gradient method, which allows us to design new merging methods that have no tractable closed-form solution like Block-Diagonal Fisher Merging and improves performance for merging methods that require regularization to find a closed-form solution like RegMean.

---

> > > > ### Author Response · Authors · 2024-02-22
> > > > **Continued response**
> > > >
> > > > > Finally, in sec 5 you don’t talk about Q or Lambda at all anymore, and instead everything is done using Z’s. If we don't even use the Q_m or L_m, why did we talk about them?
> > > >
> > > > We mention $Q_m$ and $\Lambda_M$ to motivate understanding what eq. (4) is doing and give a unified framework for understanding various merging methods. The objectives defined by various $Q_m$ give rise to various $Z_m$, which can then be optimized over via the conjugate gradient method. Specifically, MaTS uses the RegMean objective for full-model fine-tuning and the block-diagonal Fisher merging objective for $(IA)^3$.
> > > >
> > > > > After eq 8 paper claims that F_m is a per-example covariance. I assume example means a single datapoint (x,y), but Fisher computes over entire dataset D_m. Maybe “example” refers to “task”? Please clarify.
> > > >
> > > > Thanks for pointing this out. We have updated the draft to mention "$F_M$ is the covariance of the per-example gradients of the datapoints from a task."
> > > >
> > > > > The Qm in Fisher case is a first-order approximation of the second-order Hessian, which is very inaccurate, especially when we talk about million/billion dimensional matrices. Why does it make sense to claim that Qm are viewed as Hessian eigenvectors: surely these are very different things?
> > > >
> > > > The (true) Fisher is equal to the Hessian in expectation when negative log likelihood loss is used [2, 3, 4]. Also, the (true) Fisher is equal to the Gauss Newton Hessian (for appropriate loss functions including cross entropy loss which we use) and the Gauss Newton Hessian is equal to the Hessian when the “loss is minimized individually for each training example”. [3, 4]
> > > >
> > > > > The discussion in sec 3.2. states that a projection QLQ will upweight important components. This is misleading. The decomposition will do this only if the underlying covariance C_m does this, and the decomposition itself doesn’t really do anything here. Hence, you should instead discuss here the specific effect of specific Cm choices, and not claim that the SVD does parameter relevance detection.
> > > >
> > > > Thanks for pointing this out. We have updated the draft to mention m: “Overall, eq. (9), when $Q_m\Lambda_mQ^T_m$ is the decomposition of a covariance matrix, can be seen as upweighting the “important” components of the model as measured in the task subspace so that the important components don’t get washed out during merging.” We mention the specific effect of various choices for  $C_m$ in section 3.1.
> > > >
> > > > > Please include an algorithm block of the overall method.
> > > >
> > > > We have added an algorithm block to the Appendix Section C on page 22.
> > > >
> > > > [1] Gabriel Ilharco, Marco Tulio Ribeiro, Mitchell Wortsman, Suchin Gururangan, Ludwig Schmidt, Hannaneh Hajishirzi, Ali Farhadi. Editing models with task arithmetic. International Conference on Learning Representations. 2023
> > > >
> > > > [2] Agustinus Kristiadi. Fisher Information Matrix.  https://agustinus.kristia.de/techblog/2018/03/11/fisher-information/
> > > >
> > > > [3] Roger Grosse. Chapter 2: Taylor approximations. Lecture notes in University of Toronto CSC2541, Topics in Machine Learning: Neural Net Training Dynamics, 2022a. https://www.cs.toronto.edu/~rgrosse/courses/csc2541_2022/readings/L02_Taylor_approximations.pdf
> > > >
> > > > [4] Roger Grosse. Chapter 3: Metrics. Lecture notes in University of Toronto CSC2541, Topics in Machine Learning: Neural Net Training Dynamics, 2022b. https://www.cs.toronto.edu/~rgrosse/courses/csc2541_2022/readings/L03_metrics.pdf.

---

> ### Author Response · Authors · 2024-02-15
> **Continued Response**
>
> >The paper frames its discussion against “task subspaces”. I do not agree with this ...
>
> Since our paper focuses on the standard and widespread setting of merging individual-task models fine-tuned from the same base model, relating a given model to a particular task is natural. While the merging methods we study could in principle be applied to any pair of models that share a common architecture whether or not each model is associated in a particular task, in practice merging is only considered in the setting of task-specific fine-tuned models. In this setting, it is natural and intuitive to relate components of the SVD to different tasks. It is not necessarily true that the posteriors "can’t come from having different tasks since then the parameter vectors would not be same size" because, as discussed above, a given model architecture may be amenable to many different tasks (for example, in the text-to-text approach used by T5 in our NLP experiments). When different tasks necessitate different classification heads, we follow past work and merge only the "body" (i.e. backbone/base model) of the model [1, 2, 4].
>
> >The paper does not acknowledge that the SVD inverse simplifies into one where the inverse is only over eigenvalues ...
>
> Note that the inverse in equation (4) is the inverse of a sum of SVDs, not an individual SVD. While this inverse can simplify in certain cases (for example, when $\mathbf{C}_m$ are diagonal matrices as in the case of Diagonal Fisher merging), it does not simplify in general. In addition, even if a simplified inverse exists, it may not be tractable because inverting a matrix via inverting the eigenvalues would still require computing the eigenvalues of the matrix. For example, assuming a 1024x1024 weight matrix, the outer product of gradients would have dimension 1M by 1M and would therefore be intractable to compute, thus requiring approximations such as the fisher approximation introduced in K-FAC. We are not aware of any method to invert a sum of block-diagonal Fishers that scales to large models except for CG when used with the Fisher approximation introduced in K-FAC.
>
> > The paper needs to include simple ensemble as a baseline ...
>
> Ensembling has been shown to help performance when combining individual models trained on the same task. Merging focuses on a different setting where the individual models were trained on different tasks. We therefore are not aware of any past work on merging that includes ensembling as a baseline, possibly because the computational costs are so high (i.e. being $M$ times more expensive than using a single merged model). However, given the simplicity of the method, we agree that it would be an interesting baseline to include and therefore have included ensembling as a baseline method. Notably, in the multitask setting, ensembling outperforms simple averaging and task vectors, but still produces dramatically lower performance than MaTS.
>
> > I don’t think Fig1 makes much sense ...
>
>  Thanks for pointing out that you found the diagram to be confusing. We have updated the diagram in figure 1 in the following ways:
> - We removed the stars and diamonds, which represented the points in parameter space after performing the projection inherent in each merging method. Instead, we simply retain the arrows which show the direction of the projection (specifically, an axis-aligned projection for Diagonal Fisher merging and a projection along the steepest directions of the loss landscape for Blockwise Fisher merging).
> - The dashed lines indicate the path taken when interpolating between the parameters of the individual models. The point on these lines that corresponds to the merged model depends on the interpolation coefficient used when merging. We agree that it would be clearer to also show an example merged model, and therefore have introduced a new marker to do so.
> We hope these changes have made the diagram clearer.
>
> > I did not understand what the MATS actually is ...
>
>  Our paper introduces a unified perspective on merging and the approach of minimizing a given merging objective using CG. MaTS is the merging technique we develop based on this perspective and approach. For parameter-efficient training, MaTS uses a block-diagonal Fisher approximation. For full-model training, MaTS only uses the covariance of the input activations to approximate the block-diagonal Fisher.
>
> > I did not understand what the paper means by tasks ...  \
> > To clarify my stance ... \
> > I’m also confused what are the tasks in the first place ... \
> > Are different models re-trainings, or different datasets, or different classes, or ... \
> > Also, why do we want to merge models in the first place? ... \
> > If the multitask seems to always be superior, why do we want to merge? ... \
>
> We answer these questions at the beginning of our response.

---

> > ### Author Response · Authors · 2024-02-15
> > **Continued response**
> >
> > > I did not understand 6.4 ...
> >
> > We add the results of the intermediate-optimized method itself (called sequential training in Table 2). We emphasize that in this experiment, we follow the same setup as previous works [2] for a fair comparison.
> >
> > > The paper mixes and and and randomly, which makes the manuscript look unfinished ... \
> > > eq (unnumbered, after eq 2 [please number all equations]) has F while it should have F_m
> >
> > Thank you for the feedback. We have updated it in the revised version.
> >
> > > The paper implicitly assumes that all models have same parameters with same meaning ...
> >
> > We agree that the assumption that the individual models share an architecture and initialization is limiting, but this is an assumption made by most past work on model merging [1, 2, 4, 19, 20, 21, 22, 23, 34].  We do explicitly state this assumption in the first sentence of the background section. We argue that assuming different models share a common initialization is not overly limiting due to the widespread use of transfer learning in many applications of machine learning.
> >
> > > Eq 1 should cite and discuss SWA(G) (Wilson et al)
> >
> > We discuss SWAG in our related work on different ways to estimate the covariance matrix for combining models. Note that the focus of SWA(G) is quite different from merging, as SWA(G) focuses on merging different checkpoints of a model trained on the same dataset from the same training run.
> >
> > > What is $1 / \sum_m F_M$? Is this an elementwise inverse, or matrix inverse ...
> >
> > This is an element-wise inverse after the diagonal approximation. The matrix inverse of a diagonal approximation is equivalent to the element-wise inverse.
> >
> > > It’s a bit strange that eq 5 does not acknowledge the obvious simplification $(QLQ^T)^{-1} = Q L^{-1} Q^T$ ...
> >
> > While $(QLQ^T)^{-1} = (Q L^{-1} Q^T)$, in eq 5 we have a $(\sum_{m=1}^M Q_mL_mQ_m)^{-1}$ where the $Q_m$ differ across models, which does not simplify in general.
> >
> > > For vision, what was the CLIP trained against ...
> >
> >  For CLIP, we use the code and trained checkpoints released in [1]. They fine-tuned the CLIP vision backbone model with a classifier head on top for each of the 8 tasks separately. During test time, we evaluate the merged model on each of 8 tasks individually and report the average performance across the 8 tasks. The images are all rescaled to the appropriate resolution for CLIP (i.e. 224 x 224), as is commonly done. The number of classes for each task differ, ranging from 10 for MNIST to 397 for SUN397. Training on the 8 labels sets together is possible, but if different people have already fine-tuned CLIP on different datasets, we can reuse the compute and merge these models to get 1 model that does well on multiple tasks.
> >
> > > What does “base model” mean in the introduction?
> >
> >  Actually, merging is incredibly widespread for Stable Diffusion XL for exactly the purpose you mentioned - averaging models specialized to different styles/objects/etc. indeed composes the capabilities and specialization of the individual models! Merging is so common in the Stable Diffusion community that virtually all user interfaces for generating images from Stable Diffusion include the option of merging specialized fine-tuned models. See, for example, https://www.youtube.com/watch?v=kPCf1tonSVM for a tutorial on merging diffusion-based image generation models to compose styles, content specializations, etc.
> >
> > > Surely the specialised models need to have considerable overlap in their domains. Can you clarify your setting?
> >
> > This does not appear to be true in practice - for example, in our CLIP experiments (which exactly follow the experimental setting and models from [1]), we consider models fine-tuned on tasks from quite different domains but nevertheless are able to produce performant multitask models. We agree that investigating when and why merging succeeds or fails is interesting for future work.

---

> > > ### Author Response · Authors · 2024-02-15
> > > **Continued response**
> > >
> > > [1] Gabriel Ilharco, Marco Tulio Ribeiro, Mitchell Wortsman, Suchin Gururangan, Ludwig Schmidt, Hannaneh Hajishirzi, Ali Farhadi. Editing models with task arithmetic. International Conference on Learning Representations. 2023
> > >
> > > [2] Michael S Matena, Colin A Raffel. Merging models with fisher-weighted averaging.
> > > Advances in Neural Information Processing Systems. 2022
> > >
> > > [3] Colin Raffel, Noam Shazeer, Adam Roberts, Katherine Lee, Sharan Narang, Michael Matena, Yanqi Zhou, Wei Li, Peter Liu. Exploring the limits of transfer learning with a unified text-to-text transformer. The Journal of Machine Learning Research. 2020
> > >
> > > [4] Prateek Yadav, Derek Tam, Leshem Chosen, Colin Raffel, Mohit Bansal. TIES-Merging: Resolving Interference When Merging Models. Advances in Neural Information Processing Systems. 2023
> > >
> > > [5] Jing Zhou, Zongyu Lin, Yanan Zheng, Jian Li, Zhilin Yang. Not All Tasks Are Born Equal: Understanding Zero-Shot Generalization. International Conference on Learning Representations. 2023
> > >
> > > [6] Victor Sanh, Albert Webson, Colin Raffel, Stephen H Bach, Lintang Sutawika, Zaid Alyafeai, Antoine Chaffin, Arnaud Stiegler, Teven Le Scao, Arun Raja, et al. Multitask prompted training enables zero-shot task generalization. International Conference on Learning Representations. 2922
> > >
> > > [7] Alec Radford, Jong Wook Kim, Chris Hallacy, Aditya Ramesh, Gabriel Goh, Sandhini Agarwal, Girish Sastry, Amanda Askell, Pamela Mishkin, Jack Clark, Gretchen Krueger, Ilya Sutskever. Learning Transferable Visual Models From Natural Language Supervision. International Conference on Machine Learning. 2021
> > >
> > > [8] Chelsea Finn, Pieter Abbeel, Sergey Levine. Model-Agnostic Meta-Learning for Fast Adaptation of Deep Networks. International Conference on Machine Learning. 2017
> > >
> > > [9] Timo Schick, Hinrich Schütze. Exploiting Cloze Questions for Few Shot Text Classification and Natural Language Inference. European Chapter of the Association for Computational Linguistics. 2021
> > >
> > > [10] Jaemin Cho, Jie Lei, Hao Tan, Mohit Bansal. Unifying Vision-and-Language Tasks via Text Generation. International Conference on Machine Learning. 2021
> > >
> > > [11] Jason Wei, Maarten Bosma, Vincent Y. Zhao, Kelvin Guu, Adams Wei Yu, Brian Lester, Nan Du, Andrew M. Dai, Quoc V. Le. Finetuned Language Models Are Zero-Shot Learners.  International Conference on Learning Representations. 2022
> > >
> > > [12] Jacob Devlin, Ming-Wei Chang, Kenton Lee, Kristina Toutanova. BERT: Pre-training of Deep Bidirectional Transformers for Language Understanding. Conference of the North American Chapter of the Association for Computational Linguistics. 2019
> > >
> > > [13] Bryan McCann, Nitish Shirish Keskar, Caiming Xiong, Richard Socher. The Natural Language Decathlon: Multitask Learning as Question Answering. arXiv preprint. 2018
> > >
> > > [14] Jiasen Lu, Vedanuj Goswami, Marcus Rohrbach, Devi Parikh, Stefan Lee. 12-in-1: Multi-Task Vision and Language Representation Learning. IEEE/CVF conference on computer vision and pattern recognition. 2020
> > >
> > > [15] Jiasen Lu, Christopher Clark, Rowan Zellers, Roozbeh Mottaghi, Aniruddha Kembhavi. Unified-IO: A Unified Model for Vision, Language, and Multi-Modal Tasks. International Conference on Learning Representations. 2023
> > >
> > > [16] Tianbao Xie, Chen Henry Wu, Peng Shi, Ruiqi Zhong, Torsten Scholak, Michihiro Yasunaga, Chien-Sheng Wu, Ming Zhong, Pengcheng Yin, Sida I. Wang, et al. UnifiedSKG: Unifying and Multi-Tasking Structured Knowledge Grounding with Text-to-Text Language Models. Conference on Empirical Methods in Natural Language Processing. 2022
> > >
> > > [17] Ting Chen, Saurabh Saxena, Lala Li, Tsung-Yi Lin, David J. Fleet, Geoffrey Hinton. A Unified Sequence Interface for Vision Tasks. Advances in Neural Information Processing Systems. 2022
> > >
> > > [18] Mustafa Shukor, Corentin Dancette, Alexandre Rame, Matthieu Cord. UnIVAL: Unified Model for Image, Video, Audio and Language Tasks. Transactions on Machine Learning Research. 2023
> > >
> > > [19] Leshem Choshen, Elad Venezian, Noam Slonim, Yoav Katz. Fusing finetuned models for better pretraining. arXiv preprint. 2022
> > >
> > > [20] Almog Gueta, Elad Venezian, Colin Raffel, Noam Slonim, Yoav Katz, Leshem Choshen. Knowledge is a Region in Weight Space for Fine-tuned Language Models. Findings of the Association for Computational Linguistics: EMNLP. 2023
> > >
> > > [21] Mitchell Wortsman, Gabriel Ilharco, Jong Wook Kim, Mike Li, Simon Kornblith, Rebecca Roelofs, Raphael Gontijo-Lopes, Hannaneh Hajishirzi, Ali Farhadi, Hongseok Namkoong, Ludwig Schmidt. Robust fine-tuning of zero-shot models. IEEE/CVF conference on computer vision and pattern recognition. 2023
> > >
> > > [22] Mitchell Wortsman, Gabriel Ilharco, Samir Yitzhak Gadre, Rebecca Roelofs, Raphael Gontijo-Lopes, Ari S. Morcos, Hongseok Namkoong, Ali Farhadi, Yair Carmon, Simon Kornblith, Ludwig Schmidt. Model soups: averaging weights of multiple fine-tuned models improves accuracy without increasing inference time. International Conference on Machine Learning. 2022

---

> > > > ### Author Response · Authors · 2024-02-15
> > > > **Continued response**
> > > >
> > > > [23] Alexandre Ramé, Kartik Ahuja, Jianyu Zhang, Matthieu Cord, Léon Bottou, David Lopez-Paz. Model Ratatouille: Recycling Diverse Models for Out-of-Distribution Generalization.
> > > > International Conference on Machine Learning. 2022
> > > >
> > > > [24] Nico Daheim, Thomas Möllenhoff, Edoardo Maria Ponti, Iryna Gurevych, Mohammad Emtiyaz Khan. Model Merging by Uncertainty-Based Gradient Matching. arXiv preprint. 2023
> > > >
> > > > [25] Yi-Lin Sung, Linjie Li, Kevin Lin, Zhe Gan, Mohit Bansal, Lijuan Wang. An Empirical Study of Multimodal Model Merging. Findings of the Association for Computational Linguistics: EMNLP. 2023
> > > >
> > > > [26] Enneng Yang, Zhenyi Wang, Li Shen, Shiwei Liu, Guibing Guo, Xingwei Wang, Dacheng Tao. AdaMerging: Adaptive Model Merging for Multi-Task Learning. arXiv preprint. 2023
> > > >
> > > > [27] Peng Ye, Chenyu Huang, Mingzhu Shen, Tao Chen, Yongqi Huang, Yuning Zhang, Wanli Ouyang. Merging Vision Transformers from Different Tasks and Domains. arXiv preprint. 2023
> > > >
> > > > [28] Anonymous Authors. Cross-Lingual Transfer with Large Language Models via Adaptive Adapter Merging. International Conference on Learning Representations Submission. 2024
> > > >
> > > > [29] Viraj Shah, Nataniel Ruiz, Forrester Cole, Erika Lu, Svetlana Lazebnik, Yuanzhen Li, Varun Jampani. ZipLoRA: Any Subject in Any Style by Effectively Merging LoRAs. arXiv preprint. 2023
> > > >
> > > > [30] George Stoica, Daniel Bolya, Jakob Bjorner, Pratik Ramesh, Taylor Hearn, Judy Hoffman. ZipIt! Merging Models from Different Tasks without Training. International Conference on Learning Representations. 2024
> > > >
> > > > [31] Sidak Pal Singh, Martin Jaggi. Model Fusion via Optimal Transport. Advances in Neural Information Processing Systems. 2020
> > > >
> > > > [32]  Samuel K. Ainsworth, Jonathan Hayase, Siddhartha Srinivasa. Git Re-Basin: Merging Models modulo Permutation Symmetries.  International Conference on Learning Representations Submission. 2023
> > > >
> > > > [33] Masanori Yamada, Tomoya Yamashita, Shin'ya Yamaguchi, Daiki Chijiwa. Revisiting Permutation Symmetry for Merging Models between Different Datasets. arXiv preprint. 2023
> > > >
> > > > [34] Xisen Jin, Xiang Ren, Daniel Preotiuc-Pietro, Pengxiang Cheng. Dataless Knowledge Fusion by Merging Weights of Language Models. International Conference on Learning Representations. 2023.

---

### Review · Reviewer_v8xF · 2024-01-30

**Summary Of Contributions:**

This paper presents a new merging approach for neuronal models. The core idea is to build upon a recent method that used the Fisher information matrix for averaging the different models’ parameters. Given that the Fisher information matrix is hard to manipulate in practice, prior work used its diagonal approximation which allows for a closed-form merging solution of parameters. The authors propose to replace the diagonal approximation with a block-diagonal one which is assumed to be more informative. As the latter doesn’t admit a closed-form solution, the authors present it as a solution of a matrix linear equation that is solved using conjugate gradient descent. The empirical results suggest a strong performance of the proposed method compared not only to the method based on the diagonal Fisher approximation but also to other methods.

**Audience:**

Yes

**Claims And Evidence:**

Yes

**Requested Changes:**

**Major**

1.	Provide the errors at the end of the CG optimization for the best-found number of iterations in all cases to understand why increasing the number of iterations leads to a drop in performance.

2.	For the unification of other methods to be meaningful, it would be desirable to show that more than 1 instance of algorithm based on intractable closed-form solution using CG is necessary. Otherwise, I would suggest reformatting the paper and skipping this unification altogether.

**Minor**

1.	I do not understand Figure 1 and what the position of the stars and squares represents. Can authors elaborate on this?
2.	p.3: averaging averaging -> remove repetition
3.	p.9: as both as both -> remove repetition
4.	p.10 in in table 1 -> remove repetition

**Strengths And Weaknesses:**

**Strengths**

1. The idea of approximating the Fisher matrix by a block-diagonal seems reasonable and novel
2. The empirical results are great
3. The discussion of the cost of the method as well as its positioning concerning the existing methods is interesting

**Weaknesses**

1. The discussion about the unification of other methods into one paradigm seems a bit unnecessary

I feel that the whole discussion in Section 3 is not very informative in the context of this paper. In particular, the authors’ contribution doesn’t draw much benefit from this section as in the end they improve the approximation of the Fisher matrix based on the prior work (based only on 1 instance of this unification) and this constitutes their major contribution. I do not see how this unification is linked to the latter contribution and why it is necessary to fit it into some general paradigm. I would suggest removing it and to rather follow the flow 1) the work of (Matena & Raffel, 2022) approximated Fisher with a diagonal matrix which is suboptimal; 2) we propose a block-diagonal approximation and tackle the optimization challenges that it poses. If the authors can propose other contributions that fit the framework of Section 3 and require the solution using CG then it would make sense to keep it. Currently, it is very hard to parse the link between Sections 3, 4 and 5. It is probably enough to jump from Section 2 to Section 5 and include the details from Section 4 there directly as explained above. I feel that this will simplify the manuscript and make it much clearer and easier to follow.

2. The drop in performance with a higher number of iterations of CG seems counterintuitive and calls for more explanations
I understand that sometimes early stopping can be beneficial but I feel that in this case varying the number of iterations of CG arbitrarily lacks rigor. It would be surprising to find a paper that tunes the number of iterations with a step size of 10 for SGD or Adam in this way (if I’m mistaken and it is common practice then kindly provide references to it). I think this part really deserves a more in-depth analysis as the whole benefit of the author’s contribution relies on using a numerical optimization scheme that in this case seems not very robust to this crucial hyperparameter (although the reviewer appreciates the authors’ honesty about this).

---

> ### Author Response · Authors · 2024-02-15
> **Response to Reviewer v8xF**
>
> Thank you for your constructive review and well-thought out suggestion. We respond to your comments inline below.
>
> >The discussion about the unification of other methods into one paradigm seems a bit unnecessary...  \
> > For the unification of other methods to be meaningful ... Otherwise, I would suggest reformatting the paper and skipping this unification altogether.
>
> We appreciate your well-thought out suggestion and initially thought about writing the paper with the flow you suggested. We included the discussion in sections 3 and 4 for two reasons. First, we find the connection between past merging methods (which are typically seen as wholly distinct and unrelated in their methodology) enlightening . Second, for merging in the full-model setting, the RegMean objective works better than Block-Diagonal Fisher merging, and thus MaTs optimizes the RegMean objective using CG in the full-model setting. This is particularly noteworthy because RegMean was originally proposed to be solved via a closed-form solution to a regularized objective (for numerical stability). By using CG we can avoid this regularization and improve performance. We thus have to include RegMean and Diagonal Fisher merging and thus unify them to motivate the MaTS framework. Would you prefer if we added a note to the paper that some of the unification of past merging methods can be skipped for readers who are primarily interested in our final framework and experimental results?
>
> > The drop in performance with a higher number of iterations of CG seems counterintuitive and calls for more explanations ... \
> > Provide the errors at the end of the CG optimization for the best-found number of iterations in all cases to understand why increasing the number of iterations leads to a drop in performance.
>
> We can understand why it might be counterintuitive that increasing the number of CG iterations can decrease performance. In most cases, we do find that the merging objective value decreases as the number of iterations increases. In some cases, we observe a small amount of noise in the merging objective, which we attribute to numerical instability, but even in these cases the objective value tends to decrease. In some cases, we do not observe the performance (e.g. accuracy) to increase as the number of CG iterations increases. In such cases, we view selecting the number of iterations as early stopping. Please see the new graph in figure 3 (pg 25 of Appendix) for examples of these kinds of behavior.
>
> > I do not understand Figure 1 and what the position of the stars and squares represents. Can authors elaborate on this?
>
> Thanks for pointing out that you found the diagram to be confusing. We have updated the diagram in the following ways:
> We removed the stars and diamonds, which represented the points in parameter space after performing the projection inherent in each merging method. Instead, we simply retain the arrows which show the direction of the projection (specifically, an axis-aligned projection for Diagonal Fisher merging and a projection along the steepest directions of the loss landscape for Blockwise Fisher merging).
> The dashed lines indicate the path taken when interpolating between the parameters of the individual models. The point on these lines that corresponds to the merged model depends on the interpolation coefficient used when merging. We agree that it would be clearer to also show an example merged model, and therefore have introduced a new marker to do so.
> We hope these changes have made the diagram clearer.
>
> > p.3: averaging averaging -> remove repetition \
> p.9: as both as both -> remove repetition \
> p.10 in in table 1 -> remove repetition
>
>
> Thank you for pointing this out. We have fixed this in the new version of the draft.

---

### Review · Reviewer_RU1S · 2024-02-12

**Summary Of Contributions:**

This paper studies model merging. The authors first recall several ways of merging models: simple averaging (Sec. 2.1), Fisher merging (Sec. 2.2), RegMean (Sec. 2.3), before unifying them under the weighted average formula (4). The different merging methods are then interpreted as a normalized average of the models projections into the "task subspace", i.e., the eigenspace of the covariance matrix chosen. The authors finally leverage the form of Eq. (4) to compute the merged model $\theta^*$ by Conjugate Gradient, i.e., by solving iteratively the quadratic problem $\theta^*$ is the solution to, rather than by closed-form. This method enables computing the average model in cases where the closed-form solution is intractable due to a costly matrix inversion. Finally, the authors empirically validate their method with extensive experiments (Sec. 6).

**Audience:**

Yes

**Claims And Evidence:**

Yes

**Requested Changes:**

**Questions**
- It is mentioned several times that the initialization of MaTS is important (last paragraph of Sec. 1, Sec. 6.7). However, as far as I understood the problem MaTS is solving is quadratic (see Appendix B), such that the initialization should not have any impact. What am I missing here?
- What could be an explanation of the fact that the empirical Fisher works better? (last paragraph of Sec. 2.2)
- In Eq. (4), I do not understand why $C_m$ needs to be a **covariance matrix**. Any symmetric matrix $S$ can be rewritten $S=VV^\top$, such that you can argue that it the covariance matrix of $V$, but is it needed that $V$ must be an identifiable random variable? What is the interpretation behind this? Also, based on this observation, can the authors think about other matrices $C_m$ that would make sense?
- What is the interpretation for having "the gradient of the output constant" in Sec. 5.1?
- Figure 1 could be improved in many ways: 1) I assume that the ellipsoids represent the level sets of the losses for two different tasks, but I feel this should be clearly specified. Also, why are models A and B not at the center of the ellipsoids in this case? Is the landscape convex, as suggested by the figure? 2) Do the directions in which the models are transformed match the discussion: BFM seems to translate the models in a direction in which the loss landscape changes quickly. Is it made on purpose on the figure? If yes, this should be recalled in the legend I guess. 3) Based on this 2D representation, which averaging method should we prefer? I assume we want small loss for both tasks, i.e., being as central as possible in both ellipsoids, such BDF seems to perform best here. Overall, even if I understand that Figure 1 is just illustrative, I feel more concrete context should be given.


\
**Organization**
- Sec. 3: I would specify that not all model merging methods can be written under the form (4). This would allow to start introducing the methods benchmarked in Sec. 6.
- Sec.3: I would bring Appendix A.3 in the core text. It is rather short and helps understanding the relationship between the matrices.
- Sec.4: I would bring Appendix B in the core text. Indeed the CG method has an important role in this work, and deserves to be discussed more, especially the motivations for choosing this precise algorithm.
- I would remove the last sentence of the first paragraph of p.7 "K-FAC additionally [...] covariance matrices". Indeed, this approximation is not used nor relevant yet, which makes it hard to understand. It is discussed again in the next paragraph, which seems enough to me.


\
**Typos**
- p. 3: averaging averaging
- p.3, second display: should be $F_m$ and not $F$ I guess
- p.3, third display: the $1/F_m$ is not a clear notation when $F_m$ is a matrix, it is better to right properly the inverse, as in Eq. (6)
- p.4: can therefore **be** viewed as forming
- p.6: since **it** is the sum of the covariance
- p.6: that **it** allows for solving linear system
- p.7: would **cause** the merged model's performance
- p.7: (IA)$^3$ is not defined not discussed
- p.9: BFM is not defined yet
- all paper: sometimes bold subscript $m$, like for $\Lambda_m$, sometimes no (for $\theta_m$, $F_m$, etc...)

**Strengths And Weaknesses:**

**Strengths**
- the paper is globally clear and well written
- unifying different merging methods under the common expression (4) is interesting
- the experimental evaluation is comprehensive

\
**Weaknesses**
- the unified view does not account for all the methods benchmarked in Sec. 6
- more generally, I feel the common expression (4) could have been exploited more, e.g., to devise new model merging techniques by using innovative covariance matrices $C_m$
- apart from the unifying expression (4), the main (methodological) contribution is only to solve a linear system by Conjugate Gradient rather than by using the closed-form, which is relatively basic and known

---

> ### Author Response · Authors · 2024-02-15
> **Response to Reviewer RU1S**
>
> Thank you for your review and good questions. We respond to your comments inline below.
>
> > the unified view does not account for all the methods benchmarked in Sec. 6
>
>  It is true that Task Arithmetic [1] (and thus also methods based off of Task Arithmetic like TIES Merging [5]) do not fit into the unified perspective we developed. However, we develop for the first time a unified perspective on many merging methods that are *not* based on Task Arithmetic (which are typically seen as wholly distinct and unrelated in their methodology). We also note that the fact that our framework allows a flexible choice of initialization makes it able to leverage merging methods for which there is no corresponding objective.
>
> > more generally, I feel the common expression (4) could have been exploited more ...
>
>  We agree that equation (4) can be used to devise new model merging techniques, such as our novel use of the block-diagonal Fisher matrix as a covariance matrix. We further agree that our framework and perspective could be used to develop new performant merging methods in future work.
>
> > apart from the unifying expression (4), the main (methodological) contribution is only to solve a linear system by Conjugate Gradient rather than by using the closed-form, which is relatively basic and known
>
>  Though solving a linear system by Conjugate Gradient is well known, viewing merging methods as solving a linear system provides a novel framing that allows for using Conjugate Gradient to solve merging objectives that would otherwise be intractable.
>
> > It is mentioned several times that the initialization of MaTS is important ...
>
> The choice of initialization affects the convergence speed of CG because we only run CG for at most 100 iterations. Assuming CG has been run long enough, the initialization is not important. We also note a given merging objective only approximates the merged model's (generalization) performance, so it may not be advantageous to find the exact solution. Given that we are not running CG to convergence and that we don't necessarily aim to find the exact solution, the initialization becomes important.
>
> > What could be an explanation of the fact that the empirical Fisher works better?
>
> In fact, neither the true or empirical Fisher dominated across the settings we considered. Consequently, we prefer to use the empirical Fisher due to its lower computational cost.
>
> > In Eq. (4), I do not understand why $C_m$ needs to be a **covariance** matrix ...
>
> The motivation for C_m being a covariance matrix of some data rather than just as VV^T is that the interpretation of eq. (4) is that is it computes a merged model by upweighting the components of the individual models along the “task subspace”, where the task subspace are the right singular vectors V of some data P. Thus, V should represent the important dimensions in parameter space for a model to solve a given task. If C is some data, then V represents the directions of highest variance of this data and therefore captures the "important" dimensions.
>
> One other matrix C_m we experimented with earlier was computing the covariance on the gradient of the output activations. It worked better than Simple Averaging and Diagonal Fisher merging [2], but not as good as RegMean [3], so we did not include it in our paper.
>
> > What is the interpretation for having "the gradient of the output constant" in Sec. 5.1?
>
> The gradient of the output activation being constant means that the different neurons in a linear layer are independent and that the different neurons in a linear layer are all equally important.

---

> > ### Author Response · Authors · 2024-02-15
> > **Continued response**
> >
> > > Figure 1 could be improved in many ways:
> >
> >  Thank you for the suggestions to improve the figure.
> > A and B not being at the center of the eliipsoids and the landscape being convex are *not* meant to be implications from our diagram, but were done to make the diagrams in the Figure simpler. We note this also follows how other works have represented loss landscapes [4].
> >
> > Yes, the ellipsoids represents the level sets of the losses for their respective tasks. We have added this to the caption. The directions in which the models are transformed match the discussion (i.e. BFM is meant to be directions in which the loss landscape changes quickly). We have added this to the caption as well.
> >
> > Yes, the averaging method to prefer is BFM since it is as central as possible in both ellipsoids and changed the diagram to reflect this.
> >
> > Based on your suggestions, we have additionally updated the diagram in the following ways:
> > We removed the stars and diamonds, which represented the points in parameter space after performing the projection inherent in each merging method. Instead, we simply retain the arrows which show the direction of the projection (specifically, an axis-aligned projection for Diagonal Fisher merging and a projection along the steepest directions of the loss landscape for Blockwise Fisher merging).
> > The dashed lines indicate the path taken when interpolating between the parameters of the individual models. The point on these lines that corresponds to the merged model depends on the interpolation coefficient used when merging. We agree that it would be clearer to also show an example merged model, and therefore have introduced a new marker to do so.
> > We hope these changes have made the diagram clearer.
> >
> > > Sec. 3: I would specify that not all model merging methods can be written under the form (4). This would allow to start introducing the methods benchmarked in Sec. 6.
> >
> > Thank you for the feedback. We updated section 6 to mention not all merging methods fit our framework.
> >
> > > Sec.3: I would bring Appendix A.3 in the core text. It is rather short and helps understanding the relationship between the matrices. \
> > > Sec.4: I would bring Appendix B in the core text. Indeed the CG method has an important role in this work, and deserves to be discussed more, especially the motivations for choosing this precise algorithm.
> >
> > We also agree that adding Appendix A.3 and Appendix B in the main text would make it more clear. However, currently, we are at the page limit and are not sure what part we can remove. We are happy to hear suggestions.
> >
> > > p.3, third display: the is not a clear notation when is a matrix, it is better to right properly the inverse, as in Eq. (6)
> >
> >  This inverse is a element-wise division, since for a diagonal approximation, the inverse is 1 over the diagonal elements.
> >
> > > Typos
> >
> > Thank you for the feedback. We have updated the text accordingly.
> >
> > [1] Gabriel Ilharco, Marco Tulio Ribeiro, Mitchell Wortsman, Suchin Gururangan, Ludwig Schmidt, Hannaneh Hajishirzi, Ali Farhadi. Editing models with task arithmetic. International Conference on Learning Representations. 2023
> >
> > [2] Michael S Matena, Colin A Raffel. Merging models with fisher-weighted averaging.
> > Advances in Neural Information Processing Systems. 2022
> >
> > [3] Xisen Jin, Xiang Ren, Daniel Preotiuc-Pietro, Pengxiang Cheng. Dataless Knowledge Fusion by Merging Weights of Language Models. International Conference on Learning Representations. 2023.
> >
> > [4] Mehrdad Farajtabar, Navid Azizan, Alex Mott, Ang Li. Orthogonal Gradient Descent for Continual Learning. International Conference on Artificial Intelligence and Statistics. 2020.
> >
> > [5] Prateek Yadav, Derek Tam, Leshem Chosen, Colin Raffel, Mohit Bansal. TIES-Merging: Resolving Interference When Merging Models. Advances in Neural Information Processing Systems. 2023

---

### Author Response · Authors · 2024-02-15
**General Response**

General Response

We thank all the reviewers for reviewing our work and giving constructive feedback and suggestions.

1. We have updated the draft as follows (in red in the updated version): We added background about the following questions to the introduction.
- What a task is?
- What is model merging?
- Why do we want to merge models?
- What assumptions does model merging make?

2. Update the diagram in Figure 1
- We removed the stars and diamonds, which represented the points in parameter space after performing the projection inherent in each merging method. Instead, we simply retain the arrows which show the direction of the projection (specifically, an axis-aligned projection for Diagonal Fisher merging and a projection along the steepest directions of the loss landscape for Blockwise Fisher merging).
- The dashed lines indicate the path taken when interpolating between the parameters of the individual models. The point on these lines that corresponds to the merged model depends on the interpolation coefficient used when merging. We agree that it would be clearer to also show an example merged model, and therefore have introduced a new marker to do so.

3. Added the ensembling baseline in Table 1 in all setups for constructing multitask models and in Table 2 for constructing intermediate-task trained models. Overall, we see ensembling outperforms Simple Averaging, but still substantially underperforms MaTS.

4. Added the intermediate-optimized baseline for Table 2 (called Sequential training) in all setups for constructing intermediate-task trained models. This provides an upper-bound on the intermediate-task trained models since it represents what happens if we first train on the intermediate task and then train on the downstream task of interest.

5. Added Figure 3 in the Appendix which shows how the number of iterations of CG affects the merged model’s performance and the final error of the linear system

---

### Decision · Action_Editor_96gU · 2024-03-07

**Recommendation:** Accept with minor revision

**Comment:**

All reviewers agree, after authors rebuttal and updates, that the method and the experiments are interesting and that the paper deserves to be published. The changes on the paper since the submission, and the reply to the reviews, address well their comments.

The only remaining concern that is shared by two reviewers and the AE is about the name of the method and its discussion.  "Task Subspaces" can indeed refer to a lot of things (input/output/feature/parameter spaces). Even if it is defined in the introduction now, it is better to have a precise name such as "Task Parameter Subspaces" both in the title and the rest of the paper. This would make the positioning of the contribution of the paper more clear for the wide TMLR audience.

Another minor suggestion that the authors can choose to take into account or not for the final version is to add "Fisher" at the beginning of the title, again to have a more descriptive title of the contributions.

The authors themselves state in their response that they would be OK with such a change in name of the methods so the AE proposes to accept the paper with minor revision.

**Audience:**

The paper proposes a novel way to perform Fisher merging of models and an algorithm (conjugate gradient) to do this. It is definitely of interest to the TMLR audience.

**Claims And Evidence:**

The reviewers agree that the claims and the new method are supported by numerical evidence.

---

> ### Author Response · Authors · 2024-03-28
> **Response to Action Editor 96gU**
>
> Thank you for reviewing our work and the helpful feedback. We have updated the camera ready and replaced all instances of task subspace with task parameter subspace and clarified the relationship between MaTS and the the Fisher in section 7.2. We didn’t add Fisher to the title since MaTS still uses the RegMean objective for full-model fine-tuning.